# Disordered-to-ordered transitions in assembly factors allow the complex II catalytic subunit to switch binding partners

Pankaj Sharma [1], Elena Maklashina[2,3], Markus Voehler [4,5], Sona Balintova[6,7], Sarka Dvorakova[6], Michal Kraus[6], Katerina Hadrava Vanova [6,8], Zuzana Nahacka[6], Renata Zobalova[6], Stepana Boukalova [6], Kristyna Cunatova [9], Tomas Mracek [9], Hans K. Ghayee[10], Karel Pacak [8], Jakub Rohlena [6], Jiri Neuzil [6,7,11,12] ✉, Gary Cecchini [2,3] ✉ & T. M. Iverson [1,5,13,14] ✉

Complex II (CII) activity controls phenomena that require crosstalk between metabolism and signaling, including neurodegeneration, cancer metabolism, immune activation, and ischemia-reperfusion injury. CII activity can be regulated at the level of assembly, a process that leverages metastable assembly intermediates. The nature of these intermediates and how CII subunits transfer between metastable complexes remains unclear. In this work, we identify metastable species containing the SDHA subunit and its assembly factors, and we assign a preferred temporal sequence of appearance of these species during CII assembly. Structures of two species show that the assembly factors undergo disordered-to-ordered transitions without the appearance of significant secondary structure. The findings identify that intrinsically disordered regions are critical in regulating CII assembly, an observation that has implications for the control of assembly in other biomolecular complexes.

Respiratory complex II (CII; succinate dehydrogenase, SDH) connects the Krebs cycle with oxidative phosphorylation, making it a master regulator of cell metabolism[1–3]. CII also controls cell fate by regulating succinate signaling, aerobic respiration, and hypoxic metabolism[1–3]. These functions affect a range of human diseases[4,5]; most notably CII activity suppresses tumor formation[6,7]. CII activity can be regulated in many ways. For example, it has long been known that substrate-level inhibition of succinate oxidation by oxaloacetate provides feedback control in the Krebs cycle[8–10]. Covalent inhibition of the same succinate-binding site by the small molecule 3-nitropropionate induces cell death in striatal neurons and leads to neurodegeneration[11,12]. Some mutations of the catalytic SDHA subunit of CII, which oxidizes succinate[4,13–15], also correlate with a neurodegenerative clinical presentation.

[1]Department of Pharmacology, Vanderbilt University, Nashville, TN 37232, USA. [2]Molecular Biology Division, San Francisco VA Health Care System, San Francisco, CA 94121, USA. [3]Department of Biochemistry & Biophysics, University of California, San Francisco, CA, 94158, USA. [4]Department of Chemistry Vanderbilt University, Nashville, TN 37232, USA. [5]Center for Structural Biology Vanderbilt University, Nashville, TN 37232, USA. [6]Institute of Biotechnology, Czech Academy of Sciences, 252 50, Prague-West, Czech Republic. [7]Faculty of Science, Charles University, 128 00, Prague 2, Czech Republic. [8]Eunice Kennedy Shriver National Institute of Child Health and Human Development, National Institutes of Health, Bethesda, MD 20814, USA. [9]Institute of Physiology, Czech Academy of Sciences, Prague 4, 142 20, Prague, Czech Republic. [10]Department of Medicine, Division of Endocrinology & Metabolism, University of Florida College of Medicine and Malcom Randall, VA Medical Center, Gainesville, FL 32608, USA. [11]School of Pharmacy and Medical Science, Griffith University, Southport, QLD 4222, Australia. [12]1st Faculty of Medicine, Charles University, 128 00, Prague 2, Czech Republic. [13]Department of Biochemistry, Vanderbilt University, Nashville, TN 37232, USA. [14]Vanderbilt Institute of Chemical Biology, Nashville, TN 37232, USA. ✉e-mail: j.neuzil@griffith.edu.au; jiri.neuzil@ibt.cas.cz; Gary.Cecchini@ucsf.edu; tina.iverson@vanderbilt.edu

Although data are limited[16–18], the synthesis of a number of findings suggests that CII activity is regulated at the level of assembly in some biological situations. This may be a relatively complex process as mature CII contains four protein subunits (SDHA, SDHB, SDHC, and SDHD) and houses five permanently-associated cofactors (covalent FAD, [2Fe-2S], [4Fe-4S], and [3Fe-4S] clusters, and integral-membrane *b* heme)[19]. Potentially to allow for rapid onset of CII activity, CII assembly intermediates appear to be stable and can accumulate in cells. For example, the SDHA subunit can accumulate with one or more of its assembly factors as a long-lived complex of ~100 kDa, also referred to as CII$_{low}$, under conditions with stalled CII assembly[20]. In addition, dynamic disassembly of mature CII induces the appearance of a ~100 kDa species containing unassembled SDHA during macrophage activation[21] and as a response to acute ischemia, such as observed in heart attack and stroke[22–24]. In each of these cases, the removal of SDHA from assembled CII is expected to affect cellular succinate oxidation, which in turn regulates succinate signaling and metabolism. Understanding the nature of these stable intermediates and how SDHA transitions between them can therefore provide insight into the shifts in metabolism that hallmark diseases and biological states associated with changes in CII activity.

In eukaryotes, CII biogenesis involves at least four SDH assembly factors (SDHAFs)[25–30]. Individuals with mutations in these SDHAFs may exhibit pathologies associated with CII insufficiency[26–28,31]. Of these, SDHAF2 and SDHAF4 both interact directly with the catalytic SDHA subunit of CII[28,32]. In conjunction with the FAD cofactor and a small molecule dicarboxylate[32,33], these assembly factors are proposed to be involved in post-folding SDHA maturation events including correct cofactor association[26,28,32,34] (Fig. 1). SDHAF2 enhances covalent FAD attachment to SDHA[26,32], but the role of SDHAF4 is somewhat cryptic. Nevertheless, loss of SDHAF4 reduces CII assembly, promotes mitophagy, induces dilated cardiopathy in mice[35], and induces neurodegeneration in drosophila[28]. Conversely, SDHAF4 appears to be downregulated as a response to pathological stress in human cardiac muscle[35].

Synthesis of past work suggests that SDHAF2 and SDHAF4 each form binary interactions with SDHA and that they likely bind in a defined temporal order. For example, SDHAF2 increases the affinity of SDHA for dicarboxylate and FAD[33] suggesting that SDHAF2 may bind before these small molecules. SDHAF2 and dicarboxylate enhance covalent flavinylation[32], while SDHAF4 binds SDHA that contains covalent FAD in yeast[28], implying that the action of SDHAF4 follows that of SDHAF2.

In this study, we combined cellular studies, structural studies, and in vitro biochemistry to identify metastable soluble species that contain SDHA, SDHAF2, and SDHAF4. Changes in the intrinsic disorder of the SDHAF2 and SDHAF4 assembly factors are observed across the different complexes, and we show that the disordered regions of these assembly factors are critical for the transfer of SDHA between complexes. Structural parallels found via retrospective analysis of other

systems suggest that changes in intrinsic disorder may guide the assembly or disassembly of complexes in many systems.

## Results

### SDHA-SDHAF2 is the first protein-protein complex formed during SDHA maturation

To interrogate the possibilities for SDHA handling, we systematically evaluated pair-wise interactions between apo-SDHA, and either SDHAF2 or SDHAF4 using pull-down assays. These assays were also designed to determine whether the cofactor status of SDHA impacts the ability of SDHA to associate with each of these assembly factors.

Using recombinant proteins and beginning with the SDHA-SDHAF2 interaction, we found that SDHAF2 bound robustly to SDHA regardless of the presence and covalent status of FAD (Fig. 2a, lanes 1–3). This suggests that SDHAF2 binding is independent of the status of the FAD cofactor. In contrast, a similar experiment showed that SDHAF4 did not bind detectably to apo-SDHA (Fig. 2b, lane 1), and only weakly associated with SDHA with non-covalent FAD (Fig. 2b, lane 2) and SDHA with covalent FAD (Fig. 2b, lane 3). We also tested if SDHAF2 and SDHAF4 can interact directly. However, as shown in Supplementary Fig. 1 (lanes 2–4), the amount of SDHAF2 in the pull-down (Supplementary Fig. 1, lanes 2–4) is equivalent to the amount pulled-down in a negative control (Supplementary Fig. 1, lane 4). This finding suggests that the two assembly factors do not directly interact.

Consistent with a very strong binding of SDHAF2 and weak or no binding of SDHAF4 to SDHA, when apo-SDHA is exposed to a mixture of SDHAF2 and SDHAF4, only SDHAF2 binding is detectable (Fig. 2c, lane 1). The presence of non-covalent FAD has no effect on this binding pattern (Fig. 2c, lane 2). Interestingly, when SDHA contains covalent FAD, the mixture of SDHAF2 and SDHAF4 showed a reverse pattern, where SDHAF4 bound robustly and the amount of bound SDHAF2 was significantly reduced (Fig. 2c, lane 3).

Placing these findings in the context of past work, which shows that SDHAF2 enhances covalent flavin attachment to SDHA and that SDHAF4 binds to flavinylated SDHA[28,32], the most likely interpretation of these pair-wise interaction studies is that the assembly factors each bind to SDHA in a defined temporal sequence. First, SDHAF2 and fumarate bind to SDHA, which stimulates the covalent attachment of FAD to SDHA[32,33]. Subsequently, SDHAF4 binds to the holo-SDHA-AF2 complex and displaces SDHAF2. An alternative interpretation is that fumarate itself affects SDHA association with the assembly factors. To test this, we pre-formed the SDHA-AF2 complex with covalent FAD and evaluated the impact of adding SDHAF4. Consistent with a mechanism where the assembly factors bind sequentially, SDHAF4 bound robustly to SDHA and significantly displaced SDHAF2 independent of the presence of fumarate (Fig. 2d).

If the in vitro work correctly predicts this sequence of events in cells, we would anticipate that cells lacking SDHAF4 would have stalled CII assembly and would accumulate the SDHA-AF2 complex. To test this, we developed duplicate SDHAF4$^{KO}$ clones (SDHAF4$^{KO36}$ and SDHAF4$^{KO81}$) in a human adrenal pheochromocytoma cell line

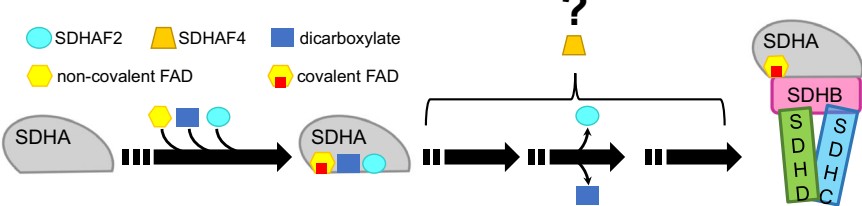

**Fig. 1 | Simplified scheme of SDHA maturation.** SDHA maturation requires FAD, dicarboxylate, SDHAF2, and SDHAF4. Each of these molecules makes binary interactions with SDHA, but the exact sequence of these binary interactions is not known. SDHA must have covalent FAD attached before it assembles into CII if the final protein complex is to be fully functional[26,74,75].

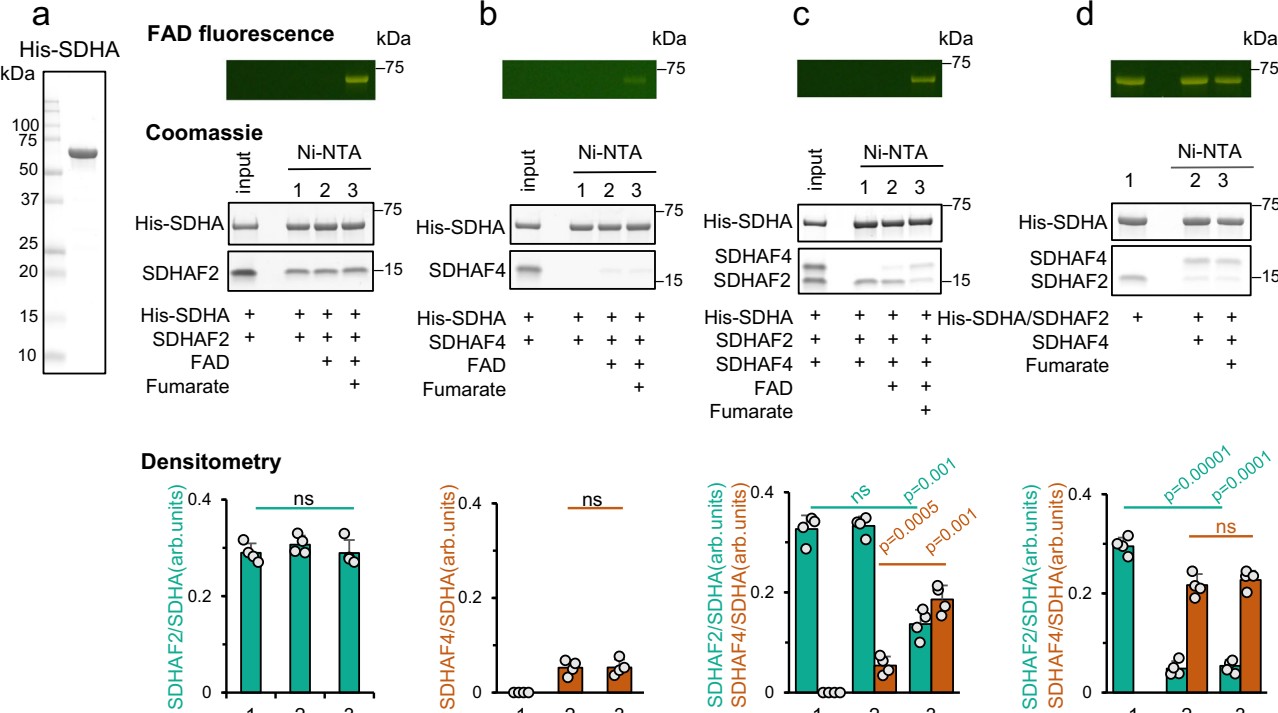

**Fig. 2 | Pairwise interactions between SDHA, SDHAF2, and SDHAF4.** The interaction of His-SDHA (2.6 μM) with SDHAF2 (6.4 μM) and SDHAF4 (8 μM) was evaluated by using a Ni-NTA pull-down assay. SDHA was tested in several of its forms: apo-SDHA, SDHA with bound non-covalent FAD, and holo-SDHA with covalently attached FAD. His$_6$-SDHA was incubated with purified SDHAF2 and SDHAF4 as indicated and associated proteins were evaluated by SDS-PAGE. FAD was added at 75 μM and fumarate was added at 5 mM. Yellowish FAD fluorescence is observed when SDHA is covalently attached to FAD. ImageJ densitometry, shown at the bottom, was measured as arbitrary units (arb. units.) and used to evaluate the relative binding of SDHAF2 (teal) and SDHAF4 (brown) to SDHA. The y-axis on the densitometry quantitation expresses these as a ratio. **a** Input protein and pairwise interaction between SDHA and SDHAF2. (left) input SDHA, (right) interaction between SDHA and SDHAF2 in the presence of FAD and fumarate. Note that only after the addition of fumarate does the covalent bond between FAD and SDHA form (lane 3). **b** Pairwise interaction of SDHA and SDHAF4. **c** Interaction of SDHA with the assembly factors after incubation with both SDHAF2 and SDHAF4. **d** Displacement of SDHAF2 after purified holo-SDHA/SDHAF2 complex (2 μM) was incubated with SDHAF4. All Coomassie gels are representative of $n = 4$ independent experiments, bar graphs show mean values ± SD, and statistics were done by paired two-tailed Student's $t$-test. Source data are provided as a Source Data file.

(hPheo1[36]) and we re-expressed SDHAF4 in SDHAF4$^{KO36}$ cells (SDHAF4$^{rec}$; see Supplementary Fig. 2a–h for characterization of the cell lines). We then evaluated whether SDHA could be assembled into CII using native blue gel electrophoresis followed by western blotting (Fig. 3).

For the parental hPheo1 cells, the vast majority of SDHA assembles into CII as expected (Fig. 3a, lane 1). In the SDHAF4$^{KO}$ cells, only a small portion of the ~72 kDa SDHA assembles into CII, while the majority now comigrates with the ~18 kDa SDHAF2 at a molecular mass of ~100 kDa (Fig. 3a, lane 2). Re-expression of SDHAF4 allowed the majority of SDHA to again assemble into CII (Fig. 3a, lane 3). Notably, SDHAF4$^{rec}$ cells had a > 10-fold increase in SDHAF4 as compared to the parental cell lines (Supplementary Fig. 2a, c) and also showed the presence of additional distinctly migrating bands that contained SDHAF4 (Fig. 3).

Given that SDHA and each of its assembly factors can comigrate at a molecular weight consistent with a complex and that these form robust direct interactions in purified proteins, one interpretation is that these proteins may be bound to each other in cells. As one step to propose the identity of the molecular species within these bands, we prepared some of the possible complexes from purified proteins, separated these on native and SDS gels, and compared the migration to the species that accumulate in cells (Supplementary Fig. 3a, b). This analysis is consistent with the assignment of SDHA-AF2 as the ~100 kDa species that accumulates in the SDHAF4$^{KO}$ cells. Although SDHA-AF4 migrates at a higher molecular weight, it is not possible to definitively assign this as the ~150 kDa species that accumulates in the SDHAF4$^{rec}$ cells (Fig. 3, lanes 2 and 3, Supplementary Fig. 2a) because other proteins could be part of the accumulating complex in cells. The remaining SDHAF4-containing bands in the SDHAF4$^{rec}$ cells resemble the laddering observed when purified SDHAF4 is separated on native gels (Supplementary Fig. 3b) consistent with these being self-oligomers. To further interrogate the ~150 kDa band, we performed an immunoprecipitation analysis of the various cell lines following transfection with FLAG-SDHA (Fig. 3b). SDHA and SDHAF4 were both present in these cell lines, but we only observe robust immunoprecipitation of SDHAF4 with SDHA in the SDHAF4$^{rec}$ cell line, which contains the ~150 kDa band.

Although the above studies suggest that these complexes form sequentially, they do not preclude a mechanism where SDHA-AF2 and SDHA-AF4 sub-assemblies form in parallel. To test this, we knocked out SDHAF2 from the SDHAF4$^{KO36}$ cells. In these SDHAF2$^{KO}$/SDHAF4$^{KO}$ cells, SDHA migrated at ~100 kDa (Fig. 3a, lane 4). We no longer detected the SDHA-AF4 species in the SDHAF2$^{KO}$/SDHAF4$^{rec}$ cells, even though these cells contained >10-fold SDHAF4 (Fig. 3a, lane 5). Instead, we observed the laddering that hallmarks the self-oligomerization of SDHAF4 (Fig. 3a, panel 4, lane 5, Supplementary Fig. 3b). This result is consistent with: (1) the in vitro studies, which showed little interaction between purified SDHA and purified SDHAF4 (Fig. 2b); (2) a robust interaction between the pre-formed SDHA-AF2 complex and SDHAF4 (Fig. 2d); and (3) the immunoprecipitation (Fig. 3b), which showed that the SDHA-AF4 complex only coprecipitated when SDHAF2 was present. Taken together, the results suggest a strong preference for the SDHA-AF2 complex to form first under normal cellular conditions and little binding of SDHAF4 to SDHA in the absence of SDHAF2.

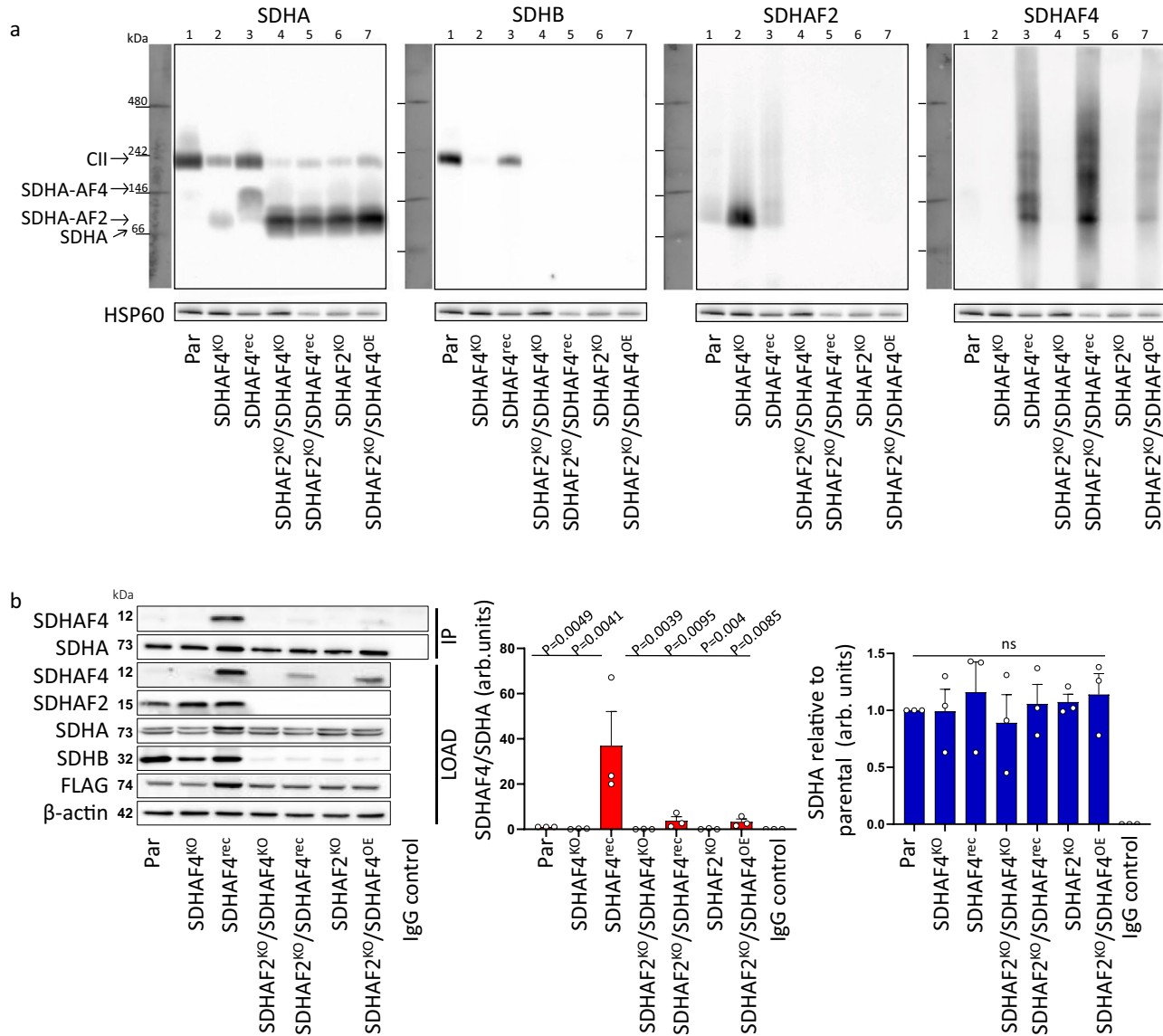

**Fig. 3 | The complexes that accumulate upon perturbation of SDHAF2 and SDHAF4 levels in cells. a** Mitochondria isolated from parental, SDHAF4[KO36], SDHAF4[rec], SDHAF2[KO]/SDHAF4[KO], and SDHAF2[KO]/SDHAF4[rec] cells were subjected to native blue gel electrophoresis followed by western blotting using anti-SDHA, anti-SDHB, anti-SDHAF2, and anti-SDHAF4 IgG. HSP60 was used as a loading control, as assessed by anti-HSP60 IgG. Positions of CII and bands positive for SDHA, SDHAF2, or SDHAF4 are indicated, with most likely possibilities for the molecular complex indicated on the left. Western blots are representative of *n* = 4 independent experiments. For SDHAF4[KO] cell lines, the quantity of CII subunits and assembly factors were generally as expected (Supplementary Fig. 2b, c); notably, the SDHAF4[rec] cells exhibited a >10-fold increase in SDHAF4 levels as compared to the parental hPheo1 cells. SDHAF4[KO] cells also showed the expected changes in routine respiration, CII-dependent respiration, CII-dependent succinate-converting activities, and succinate-fumarate ratio (Supplementary Fig. 2d–h). **b** Parental, SDHAF4[KO36], SDHAF4[rec], SDHAF2[KO]/SDHAF4[KO], SDHAF2[KO]/SDHAF4[rec], SDHAF2[KO],

and SDHAF2[KO]/SDHAF4[OE] cells were transduced for stable expression of SDHA-FLAG. Lysates from all cell lines were then subjected to immunoprecipitation using anti-FLAG IgG and analyzed by western blotting for identification of SDHAF4 binding to SDHA. Results are representative of *n* = 3 independent IP experiments from 3 separate biological replicates ± SEM. ImageJ quantitation of each species is expressed as arbitrary units (arb. units). Statistics were done using GraphPad Prism8, 2-way ANOVA, and Tukey´s multiple comparison test, with individual variances computed for each comparison. Statistical differences for the SDHAF4/SDHA values are indicated by the symbol *p* values (Par vs. SDHAF4[rec], *p* = 0.0049; SDHAF4[KO] vs. SDHAF4[rec], *p* = 0.0041; SDHAF4[rec] vs. SDHAF2[KO]/SDHAF4[KO], *p* = 0.0039; SDHAF4[rec] vs. SDHAF2[KO]/SDHAF4[rec], *p* = 0.0095; SDHAF4[rec] vs. SDHAF2[KO], *p* = 0.004; SDHAF4[rec] vs. SDHAF2[KO]/SDHAF4[OE], *p* = 0.0085). No statistical significance (ns) was found between cell lines for data on SDHA level relative to parental cells (*p* values are 0.0 to 0.99). Source data are provided as a Source Data file.

## SDHA-AF2-AF4 is the second protein-protein complex formed during SDHA maturation

In the SDHA-AF2 complex, SDHAF2 interacts with SDHA at the same surface where SDHB would bind to SDHA in the assembled CII[32]. Thus, SDHAF2 must be removed from SDHA for CII to assemble. However, displacing SDHFA2 during maturation may be non-trivial. The purified SDHA-AF2 complex is highly stable and requires denaturants for it to

disassociate in vitro[37]. To better understand how SDHAF4 displaces SDHAF2 from SDHA, we used a structural approach. We first sought to capture early intermediates in the process. To do this, we identified that temperature could affect the kinetics of SDHAF2 displacement from SDHA, with lower temperatures slowing the process. As a result, mixing SDHA-AF2 with SDHAF4 at 4 °C slows SDHAF2 displacement and results in the capture of an SDHA-AF2-AF4 intermediate.

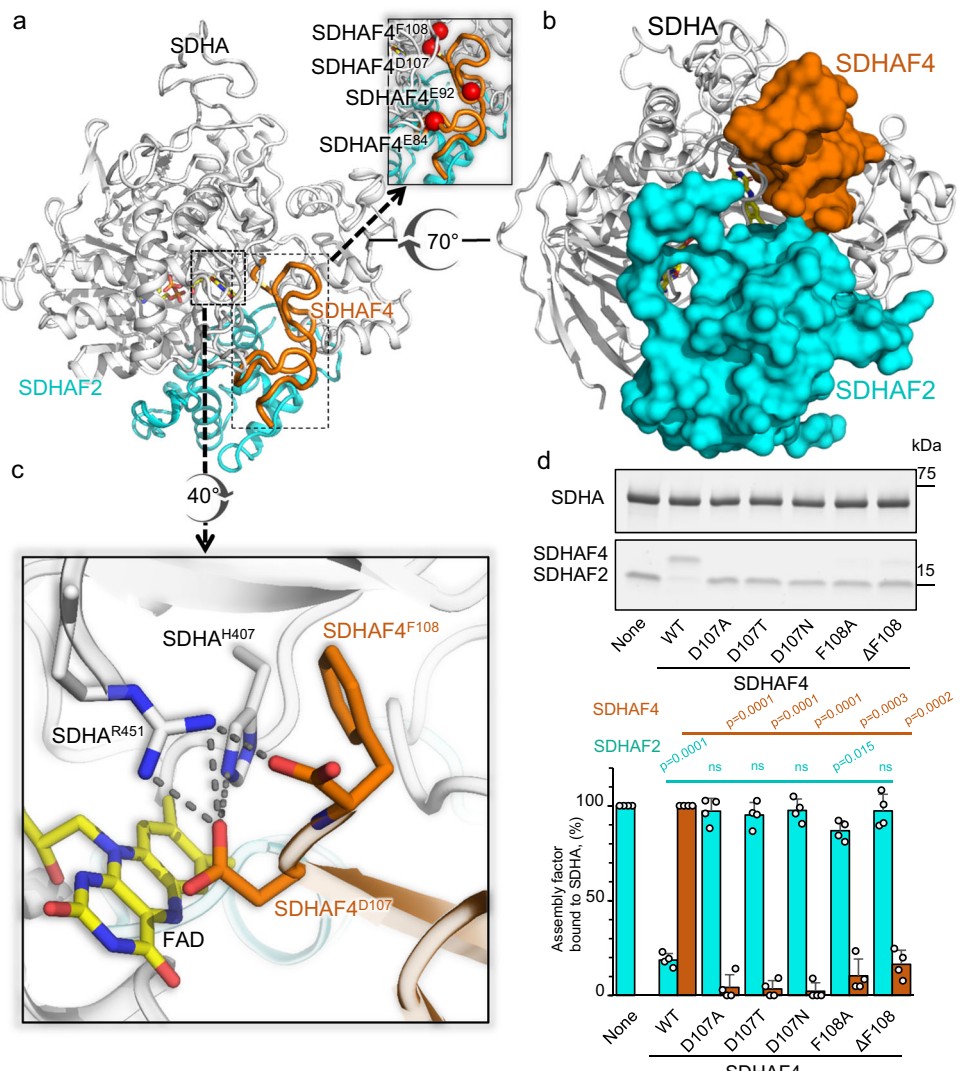

**Fig. 4 | Structure of the SDHA-AF2-AF4 complex. a** Ribbon diagram of the human SDHA-AF2-AF4 complex. SDHA is shown in gray, SDHAF2 is shown in cyan, and SDHAF4 is shown in orange. The covalent FAD is shown as a stick representation with carbons yellow, oxygens red, nitrogens blue, and phosphorous orange. The isoalloxazine functional group of the FAD is positioned between the flavin-binding domain of SDHA and the C-terminus of SDHAF4. Key interactions are shown in the inset. **b** Orientation of SDHAF2 and SDHAF4 in the complex. SDHA is shown as ribbons and SDHAF2 and SDHAF4 are shown as space-filling. The view is rotated 70° around the x-axis as compared to the view in (**a**). **c** Interactions between the C-terminus of SDHAF4 and the SDHA active site. The position of the conserved C-terminus is stabilized by interactions between SDHAF4[D107] and SDHAF4[F108] and SDHA active site residues SDHA[R451] and SDHA[H407]. **d** Validation of SDHAF4 binding residues using mutagenesis. SDHAF4 containing the indicated C-terminal mutations was evaluated for the ability to displace SDHAF2 from the SDHA-AF2 complex. The assembly factors that remained bound to SDHA were visualized after the separation of the reaction on an SDS-PAGE gel. Mutations involved SDHAF4[D107] (SDHAF4[D107A], SDHAF4[D107T] and SDHAF4[D107N]), and SDHAF4[F108] (SDHAF4[F108A] and SDHAF4[ΔF108]). The SDS-PAGE gel is representative of *n* = 4 independent experiments. ImageJ quantitation of SDHAF2 (teal) and SDHAF4 (brown) was used to calculate the percentage of each assembly bound to SDHA, as compared to a control (100%). This is expressed on the y-axis of each bar graph as mean values ± SD. Bar graphs show mean values ± SD, and statistics were calculated by paired two-tailed Student's t-test. Source data are provided as a Source Data file.

Supporting this intermediate as relevant to SDHAF2 displacement, incubating the SDHA-AF2-AF4 at 25 °C overnight or 37 °C for 4–5 h allowed for the spontaneous release of SDHAF2.

We purified the metastable species that formed from a mixture of SDHA-AF2 and SDHAF4 at 4 °C, crystallized the species, and determined the structure by X-ray crystallography (Supplementary Table 1, Fig. 4). This identified an SDHA-AF2-AF4 assembly intermediate (Fig. 4a). Here, SDHAF2 and SDHAF4 bind adjacent to each other on the surface of SDHA (Fig. 4b). Despite adjacent binding, each of these assembly factors interacts with SDHA, but not with each other (Supplementary Table 2).

As relates to the control of maturation, the most striking aspect of the SDHA-AF2-AF4 structure is the decrease in the intrinsic disorder of both assembly factors. In terms of SDHAF2, the N- and

C-termini were fully disordered in the NMR structure of the isolated yeast SDHAF2 homolog[38] and were partially disordered in the crystal structure of the SDHA-AF2 complex[32] (Supplementary Fig. 4a, b). In the SDHA-AF2-AF4 structure, both the N- and C-termini of SDHAF2 strongly interact with SDHA (Supplementary Fig. 4c, d, Supplementary Table 2), but the additional ordering does not involve a significant increase in secondary structure. Outside of the increased ordering of the termini, the fold of SDHAF2 largely resembles that described previously[32] except for a shift of amino acids SDHAF2[146-153] by 4.5 Å (Supplementary Fig. 4d).

Perhaps more intriguing is the structural organization of SDHAF4. Here, we used both circular dichroism (CD) spectroscopy (Supplementary Fig. 5a, b) followed by NMR-based backbone assignments of chromatographically pure protein (Supplementary Fig. 5c–f,

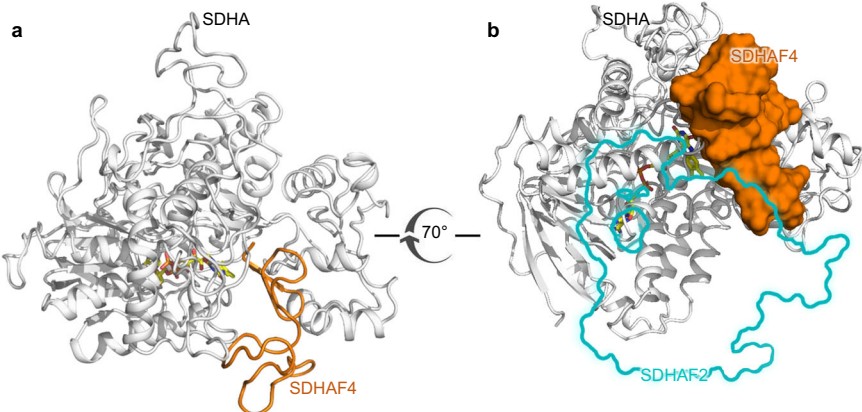

**Fig. 5 | Structure of the SDHA-AF4 complex. a** The SDHA-AF4 structure is shown as a ribbon representation with SDHA shown in gray and SDHAF4 shown in orange. Covalently bound FAD is shown in stick representation (**b**) A rotated view of the SDHA-AF4 structure with SDHAF4 shown as space-filling. The location of SDHAF2 from the SDHA-AF2-AF4 structure is shown as a cyan outline. Overlap between the binding sites for SDHAF2 and SDHAF4 suggests that the ordering of SDHAF4 could help to displace SDHAF2.

Supplementary Table 3) to show that isolated SDHAF4 is a disordered protein (Supplementary Fig. 5g, h). However, when in complex with SDHA, the C-terminal 34 residues of SDHAF4 are associated with clear electron density (Supplementary Fig. 6a). The ordered region of the protein, comprising SDHAF4$^{75-108}$, has little secondary structure and contains only two short β-strands (Supplementary Fig. 6a, b). Extensive interactions between SDHAF4 and SDHA (Supplementary Table 2) involve strongly conserved residues from both proteins and hinge upon intimate contacts between the two C-terminal residues of SDHAF4 (SDHAF4$^{D107}$ and SDHAF4$^{F108}$) and the active site SDHA$^{H407}$ and SDHA$^{R451}$ of SDHA (Fig. 4c).

Curiously, SDHAF4$^{D107}$ and SDHAF4$^{F108}$ at the C-terminus of SDHAF4 form tight interactions at the active site, with SDHAF4$^{D107}$ acting as a dicarboxylate mimetic. To do this, SDHAF4$^{D107}$ makes strong interactions with the active site residues that normally bind dicarboxylate: SDHA$^{H407}$ and SDHA$^{R451}$ (Fig. 4c). Moreover, the C-terminal SDHAF4$^{F108}$ occludes the location that would be occupied by the dicarboxylate substrate in assembled CII. Taken together, these two C-terminal residues of SDHAF4 contribute to the strong SDHAF4 binding to SDHA. In addition, these residues almost certainly preclude the binding of substrates or inhibitors to the SDHA subunit during the CII assembly process. These interactions also prevent solvent exposure of the FAD during assembly, which could prevent catalysis and impede the formation of reactive oxygen species.

Supporting the C-terminus of SDHAF4 as important for function, residues equivalent to SDHAF4$^{D107}$ and SDHAF4$^{F108}$ are strictly conserved in all SDHAF4 homologs, including putative homologs from a variety of α- and γ-proteobacteria (Supplementary Fig. 6c). To validate the importance of this interaction, we prepared five mutations of the C-terminus (Fig. 4a (*inset*)) and assessed the impact on binding and disassociation of the holo-SDHA-AF2 complex. All five variants showed reduced, albeit detectable, binding to SDHA-AF2 (Fig. 4d, Supplementary Fig. 7a–g) but were unable to disassociate the SDHA-AF2 complex (Fig. 4d).

Synthesis of the structural and biochemical data also suggests how SDHAF4 interacts with the SDHA-AF2 complex. The SDHA-AF2-AF4 structure suggests that dicarboxylate bound at the active site would compete for the binding of SDHAF4 to SDHA since SDHAF4$^{D107}$ binds at the same location and engages some of the same amino acid residues as the dicarboxylate. Our in vitro experiments (Fig. 2d), however, show that fumarate does not compete with SDHAF4 for binding to SDHA. This suggests that SDHAF4 may initiate attachment to SDHA at a site distant from the active site. Other notable residues of SDHAF4 that could initiate contact with SDHA-AF2 involve the conserved SDHAF4$^{E84}$ and SDHAF4$^{E92}$, which interact with the conserved SDHA$^{R171}$ (Fig. 4a (inset), Supplementary Fig. 7h). We again used mutagenesis to assess how these conserved residues affected the binding and disassociation of the holo-SDHA-AF2 complex (Fig. 4a (inset), Supplementary Fig. 7h–j). Here, SDHAF4$^{E92N}$ still binds to SDHA-AF2 but displaces only ~40–50% of SDHAF2 (Supplementary Fig. 7k). However, SDHAF4$^{E84N}$ did not detectably bind to the SDHA-AF2 complex (Supplementary Fig. 7k). This may suggest that the initial attachment of SDHAF4 to SDHA is through these residues more distant from the conserved SDHAF4 C-terminus. Binding would restrict the movement of the capping domain and would allow the C-terminus of SDHAF4 to fold into the location seen in the SDHA-AF2-AF4 structure.

## SDHA-AF4 is the third protein-protein complex in SDHA maturation

As the studies in cells suggest that the next metastable intermediate in the biogenesis of CII likely involves the removal of SDHAF2 from SDHA (Fig. 3, lane 3), and the studies in vitro identify that SDHAF4 can displace SDHAF2 (Fig. 2d), we hypothesized that an SDHA-AF4 complex would be the next intermediate and sought to capture this complex. To do this, we added an excess of SDHAF4 to SDHA-AF2, incubated this mixture at room temperature overnight, purified the resultant SDHA- and SDHAF4-containing species, and determined the crystal structure (Supplementary Table 1). In the SDHA-AF4 complex (Fig. 5a, b, Supplementary Table 4), nine additional amino acids of SDHAF4 are ordered as compared to SDHA-AF2-AF4, such that residues 66–108 are now associated with clear electron density (Supplementary Fig. 8a, b). Although this is accompanied by a small increase in the number of interactions between SDHA and SDHAF4 (Supplementary Table 4) most of the direct interactions appear similar to those observed in the SDHA-AF2-AF4 complex, including the intimate interactions between the SDHAF4 C-terminus and the SDHA active site (Fig. 4c, Supplementary Fig. 8c). Importantly, this newly ordered region could affect other binding partners in multiple ways. First, it would sterically conflict with the binding position of SDHAF2 that is observed in both the SDHA-AF2 and SDHA-AF2-AF4 complexes (Fig. 5b, Supplementary Fig. 8d) and with the binding position for SDHB in assembled CII (Supplementary Fig. 9). In addition, the changes in disorder of SDHAF2 and SDHAF4 allosterically affect the angle between two domains of SDHA (Supplementary Fig. 10), which modifies this same binding surface (Supplementary Fig. 9). This suggests that the change in the disorder of the assembly factors accompanies the transfer of SDHA from one binding partner to the next.

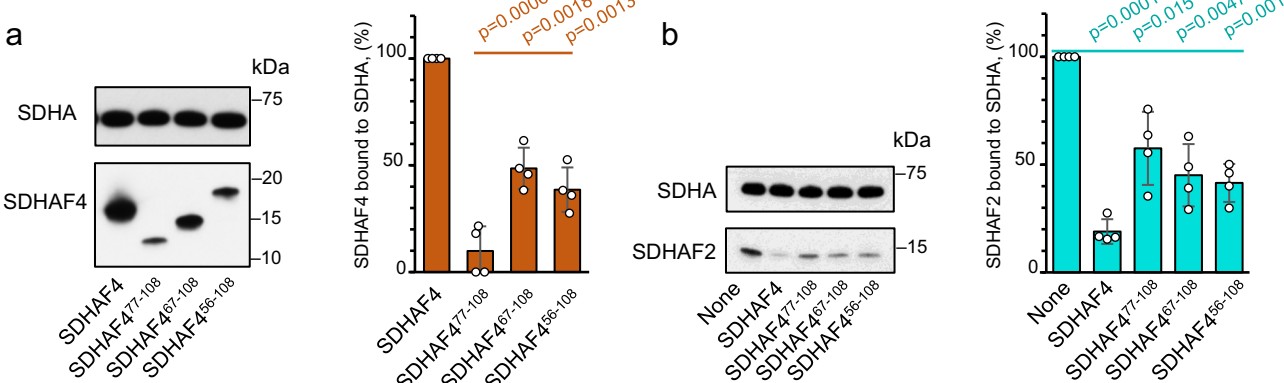

**Fig. 6 | Intrinsically disordered regions of the SDHAF4 sequence support binding to the SDHA-AF2 complex. a** Three N-terminal truncated versions of SDHAF4 (SDHAF4$^{77\text{-}108}$, SDHAF4$^{67\text{-}108}$, and SDHAF4$^{56\text{-}108}$) were designed based on prior structural[32] and cellular[28] studies and assessed for binding to SDHA-AF2. Full-length SDHAF4 (left lane) bound strongly, while each truncation variant bound at lower levels. ImageJ quantitation of SDHAF4 (brown) bound to SDHA for each mutant was measured and normalized to the amount of wild-type SDHAF4 that binds to SDHA under the same conditions. This is shown on the bar graph, with the y-axis indicating the normalized binding of SDHAF4 as a percent. **b** Regions of the protein that are intrinsically disordered facilitate SDHAF2 displacement. Each of the three SDHAF4 truncations was assessed for the ability to displace SDHAF2 from the SDHA-AF2 complex. The bar graph shows ImageJ quantitation of SDHAF2 (teal) normalized to the amount of wild-type SDHAF2 bound to SDHA, with the y-axis showing this as a percent. The SDS-PAGE gels in both (**a**) and (**b**) are representative of n = 4 independent experiments. ImageJ quantitation of each species was used to calculate the percentage bound, as expressed on the y-axis of each bar graph as mean values ± SD. Statistics were calculated by paired two-tailed Student's t-test. Source data are provided as a Source Data file.

## Intrinsic disorder of the assembly factors is required to transfer SDHA between complexes

To test whether the regions of intrinsic disorder are important for the transfer of SDHA from SDHA-AF2 to SDHA-AF4, we developed variants that removed different amounts of the disordered N-terminus of SDHAF4 and tested these for binding to SDHA-AF2 (Fig. 6a) and displacement of SDHAF2 from that complex (Fig. 6b). Of these, SDHAF4$^{77\text{-}108}$ contains only the amino acid residues that are structured in the SDHA-AF2-AF4 crystal structure. SDHAF4$^{67\text{-}108}$ and SDHAF4$^{56\text{-}108}$ contain all residues that are ordered in both crystal structures. The deleted regions of these latter two variants have no detectable sequence similarity across known SDHAF4 homologs (Supplementary Fig. 6c), further supporting that we are only eliminating regions that are always disordered.

Full-length SDHAF4 displaced ~80% of SDHAF2 from the SDHA-AF2 complex (Figs. 2d and 6b). In comparison, each of the truncations was impaired in the ability to bind to the SDHA-AF2 complex (Fig. 6a) and to disassociate this complex (Fig. 6b). The shorter constructs each removed only ~50% of SDHAF2 from SDHAF4 (Fig. 6b) despite the presence of all residues observed in the crystal structures (Fig. 5). These findings implicate the intrinsically disordered regions in the process of replacing SDHAF2 with SDHAF4 during SDHA maturation.

## Discussion

CII activity connects the Krebs cycle[39] with bioenergetics during aerobic respiration, oxygen limitation[40], or sulfide signaling[41]. This activity makes CII a master controller of metabolism. CII activity also affects succinate signaling, which has broad implications for cells. One way to regulate CII-dependent succinate oxidation is via its assembly. A critical aspect of the assembly process is the maturation and handling of the catalytic SDHA subunit.

We show that during the preferred pathway of maturation, SDHA proceeds through at least three metastable intermediates (Fig. 7): SDHA-AF2[32] (Fig. 3a, lane 2), SDHA-AF2-AF4 (Fig. 4a, b), and SDHA-AF4 (Fig. 3a, lane 3, Fig. 5a). Of these, both SDHA and SDHA-AF2 migrate as a ~100 kDa band on native PAGE. Past reports identify that there is a specific increase in the ~100 kDa species under a variety of physiological conditions. The earliest reports identified that SDHA migrated as a ~100 kDa species following mitochondrial acidification[23]. At the time, this was interpreted as an SDHA-SDHB heterodimer[23], and the concept of dynamic disintegration of CII into an SDHAB heterodimer has been retained in the literature. However, this ~100 kDa species was not evaluated for the presence of SDHB. Similar ~100 kDa assemblies containing SDHA were noted in cells where the assembly of CII was disrupted, and these were termed CII$_{low}$[20]. Mass spectrometry of a band excised from native gels detected SDHA, SDHAF2, and SDHAF4[20], although the abundance of each of these components and the nature of the underlying molecular species that formed this band was not clear at that time. Most recently, macrophage activation was shown to be associated with CII disassembly and with the appearance of a ~100 kDa species containing SDHA but not SDHB[21]. This band was not probed for the presence of assembly factors and its components beyond SDHA were not suggested. Taken together, the precise molecular composition of the ~100 kD band that accumulates under a variety of biological and pathological conditions has been elusive.

SDHAF4 knock-out cells show the accumulation of a ~100 kDa band, which may indicate that dynamic disassembly and assembly are reversible processes of each other. To determine the molecular nature of the ~100 kDa species, we combine six observations. First, western analysis of the ~100 kDa species shows that this almost exclusively contains SDHA and SDHAF2. Second, purified SDHA-AF2 migrates in a band centered at ~100 kDa (Fig. 3, Supplementary Fig. 3). Third, SDHA-AF2 is highly stable, and the purified protein does not disassociate in the absence of denaturation. Fourth, purified SDHAF4 forms oligomers, with one of these coincidentally migrating at ~100 kDa (Supplementary Fig. 3). Fifth, SDHAF4 is not consistently detected in the ~100 kDa band[20]. Sixth, both SDHA-AF2-AF4 and SDHA-AF4 migrate distinctly in bands centered at ~150 kDa (Supplementary Fig. 3, Fig. 3). We therefore propose that the molecular species that most increases in abundance under pathological conditions as a ~100 kDa band is likely the SDHA-AF2 assembly intermediate and that the inconsistent observation of SDHAF4 is due to comigration of the SDHAF4 oligomers that form under some cellular conditions. We cannot exclude the fact that isolated SDHA or other SDHA-containing assemblies are also present under some pathological conditions.

We further find that SDHAF4 allows SDHAF2 to be released from the SDHA-AF2 complex and that changes in intrinsic disorder may be important for allowing this to proceed. Mechanistically, these changes in disorder induce allosteric structural changes in SDHA

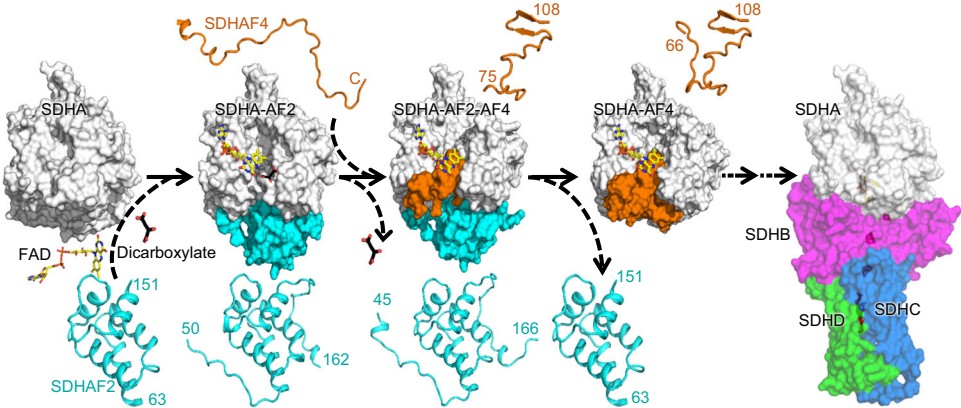

**Fig. 7 | Model for the preferred maturation pathway of SDHA.** Synthesis of the data suggests a preferred pathway for SDHA maturation in healthy cells. The compositions of SDHA-containing assembly intermediates that are formed during the CII biogenesis are shown as space-filling models in the center. Both the FAD and the dicarboxylate are shown superposed above each structure to help illustrate the maturation process, but these molecules are buried and shielded from solvent in the SDHA-AF2-AF4 and SDHA-AF4 structures. A ribbon diagram of SDHAF2 in each structure is shown at the bottom, and a diagram of SDHAF4 in each structure is shown at the top. In the first step, apo-SDHA binds FAD, dicarboxylate, and SDHAF2 to form the first metastable intermediate. In this process, the disordered termini of isolated SDHAF2 partially organize along the surface of SDHA but do not form secondary structure. This metastable SDHA-AF2 uses bound dicarboxylate as a cofactor to promote the covalent attachment of FAD, and form holo-SDHA-AF2. In the second step, SDHAF4 binds to form SDHA-AF2-AF4. Here, both termini of SDHAF2 further increase their organization. In addition, the C-terminal region of SDHAF4 becomes organized but has little secondary structure. Notably, the C-terminus of SDHAF4 occludes the dicarboxylate binding site, and dicarboxylate is released. In the third step, the additional organization of SDHAF4 displaces SDHAF2 and forms SDHA-AF4. The active site dicarboxylate-binding position remains occluded in this structure, although we note that the dicarboxylate malonate binds at a distal position in the structure. This distal site is of unknown relevance to the CII maturation process. Whether SHAF4 binds to another protein prior to or during its interaction with SDHA in cells remains unknown at this time. In addition, the handling steps of SDHA following this initial maturation process but prior to integration into functional CII (PDB ID 3SFD)[76] remains unknown at this time.

(Supplementary Fig. 10) that affect the binding surface that sequentially interacts with SDHAF2, SDHAF4, and SDHB (Fig. 7, Supplementary Fig. 9). In SDHAF2, the change in intrinsic disorder is found at the extremes of both the N- and C-termini (Supplementary Fig. 4a, b), and these regions are critical for SDHA maturation[32]. For SDHAF4, the isolated protein is fully disordered and a large segment at the N-terminus remains disordered even after binding to SDHA (Supplementary Figs. 5, 6a, b, 8a, b). The use of protein disorder to select binding partners in a temporal order may allow the system to remain dynamic. A role in protein assembly adds to the known functions of protein disorder.

Evaluation of deposited PDB coordinates identifies that changes in protein disorder accompany the biogenesis of many types of macromolecular complexes. These complexes range from small homomeric oligomers to large heteromeric complexes, include stable and transient complexes, proteins from all kingdoms, and proteins that localize in different compartments of the cell. Moreover, the changes in disorder can involve either assembly factors or the assembling protein.

Changes in disorder in assembly factors appear to be common, as exemplified by the assembly chaperone pICln (PDB entries 4F7U[42] and 1ZYI[43]) (Supplementary Fig. 11a–c) and the Cox17 and Sco1 assembly factors (PDB entries 1U96, 1U97, 2GT5, 2GQK and 2GQM) (Supplementary Fig. 11d–g)[44,45]. Changes in disorder are also observed in self-assembling complexes, as exemplified in many of the transient complexes of mammalian signal transduction. Notable exemplars are the mammalian arrestins, which can be found as monomers (PDB entry 3P2D[46]), as isoform-specific trimers (PDB entry 5TV1[47]), or in complex with each of >800 receptors[48,49]. Arrestins contain a loop that undergoes dynamic transitions during this process and can be disordered or ordered with different secondary structures, allowing the necessary plasticity to assemble into multiple types of distinct complexes (Supplementary Fig. 12a–c). A second example of changes in disorder regulating self-assembly is found in the bacteriophage nuclear shell[50,51]. Dynamics and disorder were observed at the N- and C-termini of the chimallin protein of this nuclear shell (Supplementary Fig. 13a, b) (PDB entry 7SQU[52]).

A common theme of changes in disorder during assembly is that newly ordered regions tend to lack substantial secondary structure. The paucity of secondary structure is observed in the newly ordered regions of SDHAF2, SDHAF4, arrestin[48,49], and chimallin[52] but is perhaps best exemplified in mammalian mitochondrial complex I (PDB entry 5XTD[53]) (Supplementary Fig. 14a). NDUFS4 and NDUFS6 are assembly factors that double as accessory subunits for complex I. Solution structures of isolated bacterial homologs of NDUFS4 and NDUSF6 (PDB entry 2JYA[54] and 2JRR[55]) show compact folds with disordered termini (Supplementary Fig. 14b–h). However, when bound as accessory subunits, these assembly factors have significant regions without secondary structure[56], which is accompanied by a change in the architecture of the folded regions (Supplementary Fig. 14b–h). Because secondary structure imparts stability, it is tempting to speculate that the low levels of secondary structure reflect a need for structural plasticity during the assembly process. This could help to support dynamic switching between binding partners.

Together, our findings here describe previously unknown steps of CII biogenesis. Our results indicate that intrinsic disorder in the assembly factors is important for sequential transition of SDHA subunit between metastable assembly intermediates. This ensures that the final complex contains fully mature SDHA and reduces the incorporation of a partially functional subunit, for example, SDHA with non-covalent FAD. Evaluation of assembly snapshots for unrelated structures suggests that changes in intrinsic disorder may be common during the process of assembly in a range of types of biological complexes. The ordering of these disordered regions correlates with regions of proteins that have unusually little secondary structure, which may further increase plasticity and allow the dynamic exchange of binding partners.

## Methods
### Plasmids
Plasmids pQE-hSDHA (non-cleavable N-terminal His-tag), pQT-SDHAF2 (TEV-cleavable His tag), and pQE-hSDHA-37-SDHAF2 for co-expression of His$_6$-SDHA and untagged SDHAF2 were used[32,37]. The original pQE-

hSDHA-37-SDHAF2 was constructed to produce a single polycistronic transcript that included His6-SDHA under the T5 promoter and untagged mature 37SDHAF2 (residues 37–166) that retained the Shine-Delgarno translational enhancer from the original pQE80L plasmid. For expression of SDHAF2 with a TEV cleavable 6xHis N-terminal tag, the BamHI- HindIII fragment encoding amino acid residues 37–166 of SDHAF2 was cloned into pST50Trc1-HISNDHFR (Addgene[57]). The DNA sequence encoding the mature SDHAF4 protein (amino acid residues 20–108) was optimized for *E. coli* expression, synthesized (Genscript), and cloned into the 5′ BamHI and 3′KpnI fragment of pST50Trc3-FLAG (Addgene[57]). The His$_6$-SDHAF4 plasmid was constructed using the BamHI-SalI fragment encoding amino acid residues 30–108 of SDHAF4, which was cloned into the pQT vector[37] to create pQT-SDHAF4. N-terminally truncated SDHAF4 variants were fused with the maltose binding protein (MBP) using the pMAL vector (Addgene[57]). The plasmids encoding the SDHAF4 variants (SDHAF4$^{D107A}$, SDHAF4$^{D107N}$, SDHAF4$^{D107T}$, SDHAF4$^{F108A}$, and SDHAF4$^{\Delta F108}$) were created by modification of the pQT-SDHAF4 parent vector (Genscript). All plasmids developed for protein expression in *E. coli* lacked the mitochondrial targeting sequence.

## Protein expression and purification for biochemical and structural studies

Proteins for both biochemical and structural studies were expressed in *E. coli* BL21(*DE3*). Bacteria transformed with the appropriate plasmid were grown to $OD_{600} = 0.5$ and protein expression was induced with 0.1 M IPTG (Fisher Scientific, Cat No. NC1105120) followed by growth for 17 h at either 17 °C or 20 °C. Cells were harvested by centrifugation and then stored at −70 °C until used.

For FLAG-SDHAF4, *E. coli* cells were suspended in 20 mM HEPES pH 7.5 supplemented with protein inhibitor cocktail (Roche, 11873580001), sonicated, and centrifuged at 100,000 g for 1 h. The clarified lysate was then loaded onto a DEAE-Sepharose column equilibrated with 20 mM HEPES pH 7.5 and the FLAG-SDHAF4 protein was eluted with 20 mM NaCl in the same buffer. This fraction was filtered through 30 kDa Amicon filters (Millipore Sigma, UFC903008), concentrated on 10 kDa Amicon (Millipore Sigma, UFC903008) filtration units and washed by diluting the concentrated protein 20 times with 20 mM HEPES pH 7.5, 10% glycerol, 1 mM DTT and concentration with a 10 kDa Amicon (Millipore Sigma, UFC903008) filtration unit.

Truncated SDHAF4 variants were expressed as His$_6$-MBP-FLAG fusions and purified using Ni$^{2+}$-affinity chromatography. Isolated fusion proteins were treated with TEV protease to cleave MBP in 50 mM Tris, pH 7.5, 0.5 mM EDTA, and 1 mM DTT for 17 h at 7 °C. The proteolytic reactions were applied to a Ni-NTA column and the flow-through fractions containing FLAG-SDHAF4 were concentrated with 3 kDa Amicon (Millipore Sigma, UFC900396) filtration unit.

Human SDHA-AF2 complex was purified using Ni$^{2+}$ affinity (HisTrap HP, Millipore Sigma, GE17-5248-01) followed by anion exchange (HiTrap DEAE Sepharose FF, Cytiva, 1751540). Final purification was done using size exclusion chromatography (Superdex S200 increase, Cytiva, 28990944) in 20 mM HEPES pH 7.5 buffer[32]. Apo-SDHA was expressed and purified as described above, except that the final size exclusion chromatography step was omitted. For binding assays, His$_6$-SDHAF2 was treated with TEV protease to remove the His$_6$-tag as described above[32].

## SDHA binding to SDHAF4 and SDHAF2

The interactions between SDHA and the SDHAF2 and SDHAF4 assembly factors were evaluated by a pull-down approach. His$_6$-apo-SDHA (2.6 μM) was incubated with SDHAF2 (6.4 μM), FLAG-SDHAF4 (8 μM) or both assembly factors in 40 mM HEPES, pH 7.5, 0.5 mM DTT for 30 min at 25 °C. FAD (75 μM) and/or 5 mM fumarate were added as indicated. Ni-NTA agarose resin (30 μl) was added and after an additional 10 min incubation, the resin was washed 3 times with 1.3 ml of 20 mM potassium phosphate, pH 7.5, 0.1 M NaCl, 0.5 mM DTT, and 25 mM imidazole. His$_6$-SDHA and any bound proteins were eluted with 2x SDS loading buffer (BioRad, 1610737) and separated by SDS-PAGE on "AnykD™" pre-cast protein gels (BioRad, 4568023). The gel was stained with InstantBlue Coomassie stain (Abcam, 50-196-3787) and analyzed by ImageJ (NIH).

To test the effect of SDHAF4 and fumarate on the stability of the holo-SDHA-AF2 complex, apo-SDHA (2.6 μM) was incubated with SDHAF2 (3.2 μM), 20 μM FAD, and 0.25 mM fumarate for 30 min at 25 °C. This results in the formation of holo-SDHA-SDHAF2 complex. To remove unbound components, the reaction mixture was concentrated on a 0.5 ml 50 kDa Amicon Ultra (Millipore Sigma, UFC905096) centrifugal filter unit and washed twice with 0.5 ml of 40 mM HEPES pH 7.5, 0.5 DTT. Holo-SDHA-SDHAF2 (2 μM) was next incubated with SDHAF4 in the presence or absence of 5 mM fumarate for 30 min at 25 °C and immobilized on Ni-NTA agarose beads, as described above. Note that FLAG-SDHAF4 (MW: 12.4 kDa) runs anomalously in the Tris/Glycine/SDS buffer at an apparent molecular weight of ~17 kDa. This forms a distinct band above SDHAF2 (MW: 15.6 kDa) allowing easy visualization in the Tris/Glycine/SDS buffer system. Because some truncated FLAG-SDHAF4 variants were similar in size to SDHAF2, the detection of SDHA, SDHAF2, and FLAG-SDHAF4 variants was done using western analysis with a Fast Western Kit, ECL substrate (Pierce, 35055), and the following primary antibodies: anti-FLAG, 1:1000 dilution (GenScript, A00187), anti-SDHAF2, 1:1000 dilution (SDH5, 45849, Cell Signaling Technology), anti-SDHA 1:2000 dilution (Cell Signaling Technology, 11998).

## Evaluation of the association between FLAG-SDHAF4 and SDHAF2

Purified SDHAF2 (8 μM) and FLAG-SDHAF4 (4 μM), or SDHAF2 alone, were incubated in 40 mM HEPES, pH 7.5, 0.5 mM DTT, for 30 min at 25 °C and then with 10 μl ANTI-FLAG-M2 Affinity gel (Millipore Sigma, A2220-1ML) for another 15 min. The gel was washed with 20 mM potassium phosphate pH 7.5, 0.1 M NaCl, 0.5 mM DTT, 25 mM imidazole. The proteins were eluted with 2x SDS loading buffer (BioRad, 1610737) and separated by SDS-PAGE. After Coomassie staining, the bands corresponding to SDHAF4 and SDHAF2 were evaluated by densitometry (ImageJ).

## Oligomerization of purified SDHAF4

To assess oligomerization of purified SDHAF4, His$_6$-SDHAF4$^{30-108}$ was purified using Ni-NTA followed by size-exclusion chromatography. The eluted samples were then concentrated to 5 mg/ml, a dilution series was made, and the samples were loaded onto 10% native and 10% SDS gels. Since the molecular weight marker, NativeMark™ (ThermoFisher Scientific, LC0725) runs anomalously, purified *E. coli* FrdA and human SDHA and the SDHA-AF2 complex were used as standards for native PAGE. The samples of the SDHA-AF2-AF4 and SDHA-AF4 complexes were prepared by dissolving their crystals in 20 mM HEPES PH 7.5.

## CD spectroscopy

Far-UV CD measurements of His$_6$-tagged human SDHAF4 and SDHA-SDHAF2 were recorded at 4 °C, 15 °C, and 25 °C on a Jasco J-1500 Spectropolarimeter with Peltier control using 1-mm Quartz cuvettes (Jasco, Tokyo). Far-UV CD spectra for His-SDHAF4 were collected at 50 μM, 100 μM, and 150 μM concentrations while spectra for the His-SDHA-SDHAF2 complex were collected at a 50 μM concentration. The CD spectra were collected in buffer containing 20 mM KF, pH 7.5, 0.5 mM DTT. Wavelength was scanned from 240 to 185 nm for His$_6$-SDHAF4 and from 250 to 195 nm for the His$_6$-SDHA-SDHAF2 complex. The scan speed was 100 nm/min with 10 accumulations and a response time of 1 sec.

## Perturbation of SDHAF4 levels in hPheo1 cell lines

For experiments in cells, a progenitor cell line previously derived from adrenal cells from a human pheochromocytoma tumor (hPheo1[36]), developed by Dr. Hans Ghayee, was maintained in RPMI medium (Sigma-Aldrich) supplemented with 10% fetal bovine serum, antibiotics, and pyruvate (1 mM), glucose (4.25 g/l) and uridine (50 mg/l) at 37 °C under 5% $CO_2$. Genomic deletion of the *SDHAF2* and *SDHAF4* genes was performed using the CRISPR/AsCas12a (also known as AsCpf1) system[58], chimeric Cas12a, and guide RNA-expression plasmid pX AsCpf1-Venus-NLS (gift from Dr. Björn Schuster, Institute of Molecular Genetics, Czech Academy of Sciences). Suitable crRNAs were identified using Crispor software (http://crispor.tefor.net/). Oligonucleotides were designed to provide the overhangs 5′-AGAT and 3′-TTTT (AAAA reverse complement) for cloning. Nucleotides contained an array of three crRNAs (SDHAF2[KO] and SDHAF4[KO36]) or one crRNA (SDHAF4[KO81]). The crRNA sequences were separated by AsCas12a direct-repeat sequence (AAT TTC TAC TCT TGT AGA T). For *SDHAF2*, the critical exon 2 was targeted by frameshifting. The *SDHAF4* gene has no critical exons, so exon 2 was targeted either using 3 crRNAs (clone 81) or by introducing a frameshift using a single crRNA (clone 36). The oligonucleotides were cloned into a plasmid cleaved by FastDigest BpiI (Thermo Fisher Scientific, FERFD1014), and the correct insertion was confirmed by colony PCR and DNA sequencing (primers for SDHAF4[forward]: CCA GGA AGG GAA ATG GCT GGA, SDHAF4[reverse]: CTT TTC GTT GAG TTA TTG GAG TCC T; SDHAF2[forward]: TTT CAG GGG GAT AGG GTC CG, SDHAF2[reverse]: AAT TAC CCG GGT GTG GTG AC). hPheo1 cells were transfected with the verified AsCas12a constructs using Lipofectamine 3000 (Thermo Fisher Scientific, L3000008), followed by single-cell sorting for Venus-positive cells into a 96-well culture plate. Clones were collected, and deletion of the targeted locus was confirmed by genomic PCR and/or by Sanger sequencing. The SDHAF2[KO]/SDHAF4[KO] cell line was developed using the same procedure but starting with verified SDHAF4[KO36] cells and knocking out SDHAF2.

To re-express SDHAF4 in SDHAF4[KO] cell lines, the pCDH-SDHAF4 vector was prepared. To do this, SDHAF4 cDNA[28] (a gift from Jared Rutter, University of Utah) was sub-cloned into the pCDH-CMV-MCS-EF1 vector, which contains a neomycin (System Biosciences, CD514B-1) selection marker. SDHAF4 was amplified and appropriate restriction sites were added using iProof High-Fidelity DNA Polymerase (Biorad, 1725301). Sequences of primers are as follows: SDHAF4[forward]: TAA GCA TCT AGA CCA CCA TGA CCC CAT CGA GGC TTCC, SDHAF4[reverse]: TGC TTA GGA TCC TTA AAA ATC AAT ACA GCG TCC TTT TCG TTC. Target vector and PCR products were digested using FastDigest enzymes BamHI and Xba1 (Thermo Fisher Scientific, FD0054 and FD0684, respectively). The cleaved materials were separated on an agarose gel (Nippon Genetics, AG02), purified with Nucleospin Gel and PCR clean-up kit (Macherey-Nagel, 740609), and ligated into the vector using T4 DNA ligase (Thermo Fisher Scientific, 15224017). The ligation reaction was transformed into One Shot Stbl3 chemically competent *E. coli* (Invitrogen, C737303). The SDHAF4 constructs were isolated from positive colonies using NucleoSpin Plasmid Mini kit (NoLid) (Macherey-Nagel, 740499) or NucleoBond Xtra Midi kit (Macherey-Nagel, 740410). Final constructs were sequenced by Sanger sequencing (Eurofins Genomics). Throughout construct development, DNA concentration was measured using the NanoDrop ND-2000 device (Thermo Fisher Scientific).

The pCDH-SDHAF4 plasmid was transduced into parental, SDHAF2[KO], SDHAF4[KO36], and SDHAF2[KO]/SDHAF4[KO] hPheo1 cells using recombinant lentiviruses obtained from calcium phosphate-transfected HEK 293 T cells using packaging plasmids psPAX2 (Addgene, 12260) and pMD2.G (Addgene, 12259) and the pCDH-SDHAF4 constructs. The medium containing lentiviral particles was harvested 48 h post-transfection, and the viral particles were precipitated using PEG-it (System Biosciences, LV810A-1). Target cells were transduced with viruses at multiplicity of infection MOI 5–10 and selected for neomycin (200 ug/ml, Sigma A1720) resistance.

## SDS-PAGE and Western analyses

Cells were harvested and washed twice with phosphate buffered saline (PBS). For lysis, the cell pellets were resuspended in RIPA (Radio-ImmunoPrecipitation Assay) buffer containing 20 mM Tris (pH 7.5), 150 mM NaCl, 1 mM EDTA, 1 mM EGTA, 1% (v/v) NP-40, 0.1% SDS and 0.5% sodium deoxycholate supplemented with protease inhibitor mix M (Serva) for 30 min with shaking on ice. Lysed cells were centrifuged at 16,000 g for 5 min at 4 °C, and the supernatant was collected for analysis.

For SDS-PAGE, after estimation of protein concentration using the Pierce BCA protein assay kit (Thermo Fisher Scientific, 23225), protein samples (40 μg) were mixed with 4 × Laemmli sample buffer (8% SDS, 260 mM Tris-HCl (pH 6.8), 40% glycerol, 200 mM DTT (Sigma), 0.01% bromophenol blue (Sigma)), boiled for 5 min, and separated on a 12% SDS-PAGE gel.

For western analyses, proteins were transferred from the SDS-PAGE gel onto a 0.45 μm pore nitrocellulose membrane (BioRad, 1620115). The membrane was blocked for 1 h with 5% non-fat milk and incubated with primary antibody in Tris-buffered saline/Tween-20 (TBS-T) supplemented with 2% non-fat milk overnight at 4 °C. The following primary antibodies were used: anti-SDHA (Abcam, ab14715), anti-SDHB (Abcam, ab14714), anti-SDHAF2 (Cell Signaling Technology, 45849), anti-SDHAF4 (Novus, NBP1-86324), and for loading controls anti-Hsp60 (Cell Signaling Technology, 12165), anti-VDAC1 (Abcam, ab15895) or anti-GAPDH (Cell Signaling Technology, 5174). All antibodies were diluted 1:1000 in 2% non-fat milk except for anti-SDHAF4 (1:500 dilution). The membranes were then treated with horseradish peroxidase-conjugated secondary antibodies (goat anti-rabbit, 170-6515, BioRad, or goat anti-mouse, 170-6516, BioRad) diluted 1:10,000 in TBS-T containing 2% non-fat milk for 1 h at room temperature. Blots were imaged (Radiance ECL or Radiance Plus, Azure Biosystems) using AzureSpot 2.0 software (Azure Biosystems).

## Isolation of mitochondria and native blue gel electrophoresis

To isolate mitochondria, cells were resuspended in the isolation buffer (250 mM sucrose, 10 mM Tris, 1 mM EDTA) and passed three times through an 8 μm tungsten carbide ball of a Balch homogenizer (Isobiotec) using a hand-driven 1 ml syringe. The homogenate was centrifuged at 800 g for 5 min at 4 °C. The supernatant containing mitochondria was collected, and then centrifuged at 3000 g for 5 min to remove additional impurities and at 10,000 g for 15 min to pellet mitochondria. Pelleted mitochondria were washed with the isolation buffer and stored at −80 °C. 10 μg digitonin-solubilized (8 g/g protein) mitochondria were mixed with the sample loading buffer (0.015 μl per 1 μg protein; 0.75 M aminocaproic acid, 50 mM Bis-Tris, 12% glycerol, 0.5 mM EDTA, 5% Coomassie Brilliant Blue G-250) and separated on 4–16% NativePAGE Novex BisTris gradient gels. Electrophoresis ran in three steps, i.e., using the blue cathode buffer (0.02% Coomassie Brilliant Blue G-250) at 35 V for 70 min, and then clear cathode buffer at 25 V overnight. Finally, the voltage was increased to 200 V for 2 h. For western blots, gels were incubated in the transfer buffer for 10 min with 0.1% SDS, and proteins were transferred to 0.2 μm PVDF (polyvinylidene difluoride) 0.2 μm membranes (Biorad). Western analysis after native gel electrophoresis used the same procedure as western analysis following SDS-PAGE (above).

## Evaluation of respiration

Respiration was evaluated in digitonin-permeabilized cells as previously described[59–61]. For routine respiration, cells were trypsinized, washed with PBS, resuspended in Mir05 medium (0.5 mM EGTA, 3 mM MgCl2, 60 mM K-lactobionate, 20 mM taurine, 10 mM KH2PO4,

110 mM sucrose, 1 g/l essentially fatty acid-free bovine serum albumin, 20 mM Hepes, pH 7.1 at 30 °C) and transferred to the chamber of Oxygraph-2k instrument (Oroboros). The respiration measurements were performed at 37 °C. After closing the chamber, routine respiration on intracellular substrates was recorded. Access of exogenously added substrates was allowed by permeabilizing the cellular plasma membrane with 5 µg digitonin per million cells. Complex II-mediated respiration was assessed by adding 0.5 µM rotenone, 10 mM succinate, 3 mM ADP and 10 µM cytochrome c together with a CII inhibitor (5 mM malonate). Uncoupled state was achieved by CCCP (carbonyl cyanide m-chlorophenyl hydrazone) titration to obtain the maximal respiratory rate. Antimycin A (2.5 µM) was added at the end of each measurement to inhibit ETC and the residual oxygen consumption after antimycin A addition was subtracted from all results to obtain mitochondria-specific rates.

### Immunoprecipitation from cell lysates

hPheo1 cells were transduced with SDHA Myc-DDK-tagged (Origene, RC200349L3) lentiviral particles, which were prepared as stated above. All cell lines were then selected for puromycin (1 mg/ml) resistance. For the immunoprecipitation, 80 µl of protein A/G resin slurry (Thermo, 53132) was washed with RIPA buffer (10 mM Tris pH 7.4, 1 mM EDTA, 150 mM NaCl, 1% Triton X-100) containing EDTA-free protease (Roche, 11873580001) and phosphatase (Thermofisher, A32957) inhibitor cocktail, and incubated with either anti-FLAG IgG, 1:100 dilution (Thermo, PA1-984B; 4 °C/2 h) or IgG control, 1:250 dilution (Cell Signaling, 3900 S) and washed 3 times with RIPA buffer as stated previously. Cell lysates from different hPheo1 sub-lines were obtained using the RIPA lysis buffer containing protease and phosphatase inhibitors. Concentration of proteins was determined using the BCA kit (Thermofisher, 23225), samples were diluted to the same concentration, and 750 µg of protein lysate was loaded on previously prepared A/G beads with bound anti-FLAG IgG (overnight at 4 °C). Samples were washed 3 times with RIPA buffer and eluted with hot (95 °C) SDS-based elution buffer (50 mM Tris pH 6.8, 20% glycerol, 4% SDS, 0.02% BMF, 200 mM DTT). Analysis of samples was performed by western blotting, as described above.

### In-gel succinate dehydrogenase (SDH) activity

To measure in-gel SDH activity, 30 µg digitonin-solubilized mitochondria were mixed with 10x sample buffer containing 50% glycerol and 0.1% Ponceau S dye and subjected to high-resolution clear native electrophoresis. Deoxycholate (0.05%) and lauryl maltoside (0.01%) were added to the cathode buffer. Electrophoresis was run on a 4–16% NativePAGE Novex Bis-Tris gel at 35 V for 70 min, then at 25 V overnight followed by 2 h at 200 V. To visualize the activity, gels with separated protein complexes were incubated for 20 min in the assay buffer containing 20 mM sodium succinate, 0.2 mM phenazine methosulfate, and 0.25% nitrotetrazolium blue in 5 mM Tris-HCl (pH 7.4). The reaction was stopped using a solution containing 50% methanol and 10% acetic acid, and gels were immediately photographed.

### Evaluation of succinate:quinone reductase activity

Cells grown in 6-well plates were washed with PBS and lysed in 25 mM $K_2HPO_4$ buffer (pH 7.4) by alternating freezing on dry ice and thawing at 42 °C. Protein lysates (25 µg) were resuspended in 100 mM $K_2HPO_4$ buffer and transferred to a 96-well plate. The lysates were then incubated in buffer containing 20 mM succinate, 2 µM antimycin A, 5 µM rotenone, 10 mM sodium cyanide, 50 mg/ml bovine serum albumin, and 0.015% w/v 2,6-dichlorophenol indophenol for 5 min. The signal intensity at 600 nm was recorded for 5 min, then 100 µM decylubiquinone was added and the reaction was recorded for another 30 min. Regression analysis of the linear portion of the curve was performed using GraphPad PRISM software (version 8.4.3).

### Quantitative real time PCR

Total RNA was obtained using the RNeasy mini kit (Qiagen, 74004). RNA concentration was measured using a NanoDrop 2000 instrument (Thermo Fisher Scientific). First strand cDNA was synthesized from 2 µg of total RNA with random hexamer primers using Revert Aid First strand cDNA Synthesis Kit (Thermo Scientific, K1621). qRT-PCR was performed using the CFX96 Touch Real-Time PCR Detection System (BioRad) with 5xHOT FIREPol Evagreen qPCR Supermix (Solis Biodyne, 08-36-0000 S). The relative quantity of cDNA was estimated by the ΔΔCT method and was normalized to β-actin. The following primers were purchased from Sigma: SDHAF4[forward]: 5´-GCA AGA TCA CCC CTT CTG TGT-3´, SDHA4F[reverse] 5´-AAT GGG AAT CCT CTG GTG CAT-3´; β-actin[forward]: 5´-CCA ACC GCG AGA AGA TGA-3´, β-actin[reverse] 5´-CCA GAG GCG TAC AGG GAT AG-3´.

### Succinate-to-fumarate ratio

Cells were grown in 12-well plates for 24 h, harvested, and washed twice with ice-cold 0.9% NaCl. Metabolites were extracted with 500 µl of 80% methanol in water. After centrifugation for 5 min at 16,000 g at 2 °C, 400 µl of the extract was transferred to a 2-ml glass vial with a screwcap, dried *in vacuo*, and re-dissolved in 100 µl of anhydrous pyridine (Sigma-Aldrich, 270970-100 ML). 30 µl of N-tert-butyldimethylsilyl-N-methyl trifluoroacetamide (Sigma-Aldrich, 394882-5 ML) was added. The content was vortexed and incubated at 70 °C for 30 min under constant shaking, after which 300 µl of hexane (VWR Chemical) was added. The silylated extract was analyzed using two-dimensional gas chromatography with mass spectrometric detection (GC×GC-MS; Pegasus 4D; LECO Corporation). The first dimension was run on an Rxi-5Sil MS column (30 m × 0.25 mm ID × 0.25 µm dF; Restek), and the second dimension was run using a BPX50 column (1–1.5 m × 0.1 mm ID × 0.1 µm dF; SGE). Operating conditions were as follows: primary oven temperature gradient of 50 °C (1 min) to 240 °C (10 °C/min steps), and to 320 °C (4 min) (20 °C/min steps); the secondary oven was set at 5 °C above the primary oven temperature; modulation period, 3 to 5 s; injection temperature, 30 °C; split-less injection mode was applied; injection volume, 1 µl; carrier gas, helium; and corrected constant flow, 1 ml/min. MS-detection parameters were as follows: electron ionization, −70 eV; transfer line temperature, 280 °C; and ion source temperature, 280 °C. ChromaTOF software (v.4.51; LECO Corporation) was used to control the instrument, and to acquire and process data.

Succinic and fumaric acids were analyzed as tert-butyl silyl derivatives, with their identities confirmed by co-elution with standards. Analytes were quantified using masses of m/z 289 (succinic acid) and m/z 287 (fumaric acid).

### Isotope labeling of SDHAF4 for NMR

Isotope labeling of His$_6$-SDHAF4 with $^{15}$N and $^{13}$C for NMR experiments was done by growing *E. coli* BL21(*DE3*) in M9 minimal media supplemented with $^{13}$C glucose and $^{15}$N ammonium chloride (Cambridge Isotopes). Protein was purified using Ni$^{2+}$ affinity chromatography followed by size exclusion chromatography on a 24 ml Superdex S200 increase column (Superdex S200 increase, Cytiva, 28990944) in 20 mM HEPES pH 7.5, 1 mM TCEP. Purified protein was exchanged in 20 mM sodium phosphate pH 6.5, 3 mM TCEP buffer using a second step of size exclusion chromatography on a Superdex S200 increase column. Protein was concentrated to 10 mg/ml using a 3 kDa Amicon filtration unit.

### NMR spectroscopy

NMR experiments were performed at 298 K on a Bruker AV-III 600.13 and 900.13 MHz spectrometer equipped with triple resonance cryogenically cooled CPTCI probes. All experiments were recorded with 9.5 mg/ml His$_6$-SDHAF4 in 550 µl H$_2$O/D$_2$O 95:5 (~1 mM), containing

20 mM sodium phosphate, 10 mM Tris (2-carboxylethyl) phosphine (TCEP), and -0.5 mM of 4,4-dimethyl-4-silapentane-1-sulfonic acid (DSS) for reference. NMR spectra were processed using Topspin 3.6 (Bruker BioSpin) and analyzed with NMRViewJ (One Moon Scientific, Inc.). Standard $^1$H-$^{15}$N HSQC experiments were used. Backbone resonance assignment required two steps. The first step measured and analyzed three sets of carbon direct detect measurements. The second step used the traditional amide-based method. The carbon direct detect experiments were the CON, hcaCOncaNCO, and hcacoNcaNCO[62]. The amide-based experiments were standard HNCO, HNcaCO, HNCA, HNcoCA, CBCAcoNH, HNCACB, and HSQC experiments[63]. Data acquisition parameters are described in Supplementary Table 3.

## NMR data analysis

Obstacles in assigning the SDHAF4 peptide included: (1) the limited signal dispersion in the absence of secondary structure; (2) the high percentage of prolines (about 10% of all residues), which broke the connectivities in the traditional assignment scheme because they lack amide protons; and (3) an apparent presence of a major and a minor form, as seen in the $^{15}$N HSQC spectrum that could not be readily separated (Supplementary Fig. 5e). Therefore, an alternative assignment method was used, where all C' and N backbone atoms were initially assigned using the carbon direct detect experiments with the highest sensitivity and resolution measured at 900 MHz and a cold carbon channel[64]. This information served as a bases for the traditional amide-based assignment. using a data set of the following 3D spectra: HNCO, HNcaCO, HNCA, HNcoCA, CBCAcoNH, HNCACB spectra[63]. The use of these two assignment strategies provided the necessary spectral resolution and continuous connectivity, allowing for 99% assignment of all N, C', $^1$HN,$^{13}$C$_\alpha$, $^{13}$C$_\beta$ atoms following the 6xHis N-terminal tag, starting at SDHAF$^{E20}$. Assignment of the N and C' atoms from SDHAF$^{H16}$ to the terminal SDHAF$^{F108}$ were complete. HN, C$_\alpha$, C$_\beta$, were fully assigned immediately following the 6xHis N-terminal tag, with the exception for SDHAF4$^{S35}$ (HN, C$_\alpha$, C$_\beta$) and SDHAF4S$^{S36}$(HN), which could not be assigned because no unambiguous peaks were found for them. Several amide peaks in the HSQC spectrum are weak with SDHAF4$^{S34}$ only being visualized when lowering the contour level almost to the noise level. All other missing assignments were expected, such as the C$_\beta$ atoms of the glycine residue, and the missing amide protons for the prolines. Given challenges in assigning spectra of intrinsically disordered proteins, the chemical shift index (CSI) analysis was performed using four programs in parallel. The first program, Talos+ (Supplementary Fig. 5h), uses the order parameter S$^2$ which shows high flexibility for most residues pointing to a low probability of secondary structure[65]. To validate these results, we repeated the analysis using chemical shift secondary structure population inference (CheSPI; Supplementary Fig. 5i)[66], Poulsen[67], and POTENCI[68]. All analyses are consistent with SDHAF4 as an intrinsically disordered protein. The assigned HSQC and CON spectra are shown in Supplementary Fig. 5e, f.

## Crystallization, data collection, and structure determination

The SDHA-AF2-AF4 complex was prepared by incubating purified holo-SDHA-SDHAF2 complex with purified His$_6$-SDHAF4 for 5 h at 4 °C at a protein concentration of 0.5 mg/ml in 20 mM HEPES pH 7.5, 1 mM DTT. The protein solution was then concentrated and subjected to size exclusion chromatography on a Superdex S200 increase column (Superdex S200 increase, Cytiva, 28990944) in the same buffer. The SDHA-AF4 complex was prepared by incubating purified holo-SDHA-AF2 with His$_6$-SDHAF4 at 25 °C overnight followed by size exclusion chromatography on a Superdex S200 increase column (GE Healthcare) in the same buffer.

The SDHA-AF2-AF4 complex was crystallized using the hanging drop vapor diffusion method mixing the complex (12 mg/ml) with reservoir solution (0.2 M ammonium citrate tribasic pH 8.0, 20% w/v

polyethylene glycol 3,350 and 1 mM DTT) at 1:1 ratio at 25 °C. The SDHA-SDHAF4 complex was crystallized by mixing proteins (14 mg/ml) at 1:1 ratio with reservoir solution containing 100 mM sodium malonate pH 4.7, 12% PEG 3350, and 1 mM DTT. For cryoprotection, the crystals of both complexes were dipped into a solution containing all components of the reservoir and a final concentration of 20% ethylene glycol prior to flash cooling in liquid nitrogen.

Diffraction data were collected at LS-CAT beamline ID-F ($\lambda = 0.97872$) equipped with MARMOSAIC 225 CCD detector. Data were collected at a temperature of 100 K for both SDHA-AF2-AF4 and SDHA-AF4 crystals. The data were indexed, integrated, and scaled using HKL2000[69]. Unit cell parameters and data collection statistics are listed in Supplementary Table 1.

## Structure determination and refinement

The structure of SDHA-AF2-AF4 was determined by molecular replacement using the Phaser subroutine[70] in Phenix[71] with the SDHA-AF2 complex (PDB entry 3VAX[32]) as the search model. SDHAF4 was modeled manually into the $|F_o| - |F_c|$ difference electron density in COOT[72]. The structure of SDHA-AF4 was determined by molecular replacement following the removal of SDHAF2 from the resultant SDHA-AF2-AF4 structure. Structures were refined by standard methods by alternating model building in COOT[72] and refinement in Phenix[71].

## Statistics and reproducibility

The statistical methods used in the data analysis are indicated in the corresponding figure legends and were selected depending on the groups being compared. This study contains cell line experiments validated by the independent creation of duplicate cell lines. Because the exact numbers of cells could not be determined, we instead normalized the assay by using loading controls. Data was excluded for these experiments when the loading control was not consistent for all groups. No statistical method was used to predetermine sample size. The experiments were not randomized, and blinding was not performed.

Structural work used standards in the field to exclude diffraction spots that were statistical outliers, as assessed by HKL2000[71]. Diffraction spots were also excluded if they were beyond the resolution cutoff, as determined by a composite evaluation of R$_{sym}$, completeness, I/σ, and CC$_{1/2}$.

## Reporting summary

Further information on research design is available in the Nature Portfolio Reporting Summary linked to this article.

# Data availability

PDB coordinates and processed diffraction data generated in this study have been deposited in the Protein Data Bank (https://wwpdb.org/) with accession codes 8DYD and 8DYE. Raw diffraction data have been deposited with SBGrid (http://data.sbgrid.org) with accession codes and direct hyperlinks of 954 and 955. NMR Chemical shift data for SDHAF4 have been deposited in the BMRB databank (http://www.bmrb.wisc.edu), retrievable under the accession number 52207. Source data are provided with this paper. Other PDB entries used in this study and their description: The human SDHA-AF2 complex, PDB entry 3VAX[32] was used for molecular replacement. Porcine mitochondrial Complex II, PDB entry 3SFD[73] was used for showing assembled complex II in Fig. 7. The following PDBs are cited in the discussion: 4F7U[42], 1ZYI[43], 1U96[44], 1U97[44], 2GT5[45], 2GQK[45], 2GQM[45], 3P2D[46], 5TV1[47], 7SQU[52], 5XTD[53], 2JYA[54], 2JRR[55]. Source data are provided with this paper.

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

## Acknowledgements

We thank EA Berry, J Hirboo, and V Segal for their comments on the manuscript during preparation. We thank F Mulder for assistance with NMR analysis using CheSPI. This research used resources of the Advanced Photon Source, a U.S. Department of Energy (DOE) Office of Science User Facility operated for the DOE Office of Science by Argonne National Laboratory under Contract No. DE-AC02-06CH11357. Use of the LS-CAT Sector 21 was supported by the Michigan Economic Development Corporation and the Michigan Technology Tri-Corridor (Grant 085P1000817). Use of the Advanced Photon Source was supported by the United States Department of Energy, Office of Basic Energy Sciences, under contract DE-AC02-06CH11357. NMR data were collected at the Vanderbilt Center for Structural Biology NMR core facility, which is supported in part by the National Science Foundation award 0922862, U.S. National Institutes of Health award S10 RR025677 and R35GM118089-04S1, and Vanderbilt University matching funds. This work was supported by National Institutes of Health award GM61606 (GC/TMI). GC is the recipient of a Senior Research Career Scientist award IK6BX004215 from the Department of Veterans Affairs. P.S. was supported by Postdoctoral Fellowship 19POST34450093 from the American Heart Association. This work was also supported in part by the Czech Science foundation awards to J.N. (20-05942 S and 21-04607X), J.R. (19-20553 S), S.B. (20-11724Y and 23-05303 S) and T.M. (20-25768 S). and a Czech Health Foundation award to S.D. (NU23-03-00226). Infrastructure support was provided by Czech Academy of Sciences for the Institute of Biotechnology (RVO86652036).

## Author contributions

Conceptualization: T.M.I., G.C., J.N., J.R., E.M., and P.S.; methodology: P.S., E.M., M.V., S.D., S. Balintova, M.K., K.H.V., S. Boukalova, H.G., R.Z., Z.N., J.R., K.C., T.M.; Funding acquisition: J.N., J.R., T.M.I., and G.C. Writing-original draft: P.S., T.M.I., E.M., G.C., and J.N. Writing-review & editing: T.M.I., G.C., J.N., J.R., P.S., E.M., M.V., S.D., S. Balintova, and K.P.

## Competing interests

The authors declare no competing interests.
