## [Peer Review File · Nature Communications]

Disordered-to-ordered transitions in assembly factors allow the complex II catalytic subunit to switch binding partnersREVIEWER COMMENTS

Reviewer #1 (Remarks to the Author):

In this manuscript “Disordered-to-ordered transitions in assembly factors allow complex II catalytic subunit to switch binding partners” by Sharma et al., the authors present the first atomic resolution structure of the human complex II assembly factor SDHAF4 in a ternary complex with the flavo-protein holo-SDHA and the assembly factor SDHAF2 and additionally in a binary complex with holo-SDHA. The ternary complex presented here shows that SDHAF2 and SDHAF4 occupy distinct binding sites on SDHA. Interestingly the conserved C-terminal residues in SDHAF4 interact directly with amino acid residues within the active site of holo-SDHA. The structures presented here (when compared to the published human SDHA-SDHAF2 complex and yeast SDHAF2) illustrate changes in intrinsic disorder of the assembly factors in the free and complexed states. In the abstract the authors state that a gain of structural order is required for release of SDHA from its assembly factors.

The team also provided data from in vitro binding studies and native gel analysis of recombinant protein complexes and an analysis of steady state protein complexes in mitochondria isolated from parental and selected knockout cell lines. They have attempted to draw conclusions from the data about the order of assembly of complex II and the composition of intermediates by comparing relative migration on separately run native gels (see further specific comments below in point 5). The analysis of the SDHAF4 knock out cell line does show clear accumulation of a low molecular weight complex (which based on relative migration is likely, but not definitively, the SDHA-SDHAF2 complex) suggesting SDHAF4 is required for progression of SDHA along the assembly pathway. The in vitro binding studies support such a role to some degree, however these data are not incontrovertible. Furthermore, the statement in the abstract that SDHAF4 must displace SDHAF2 for assembly to proceed is poorly supported by the data presented. A significant amount of fully assembled SDHA is seen in the SDHAF4 knock out cell line (Figure 2b, panel 1, lane 2). Rather, assembly of SDHA into the terminal complex is more efficient when SDHAF4 is present in mitochondria but SDHAF4 is not essential. The rate of dissociation of SDHAF2 from SDHA is potentially enhanced via the action of SDHAF4.

The following points must be addressed by the authors.

1. Figure 2a is not labelled sufficiently. It is unclear which experiment is shown in which lane due to the lack of information provided in the figure legend, main text or through labelling. The information can only be assumed from the key and figure legend on Figure S1. The authors should provide appropriate labelling for Figure 2a so it can be critically analysed. The authors also provide a quantitation of Figure 2a on Figure S1, however here the figure shows error bars, although the figure legend implies it is from the quantitation of a single gel (Figure 2a). Does this quantitation represent technical replicates of a single experiment (N = 1), or alternatively from 4 or 5 individual experiments (N = 4 or 5) as suggested by the number of points on the graph. Related to the quantitation on Figure S1a, how was the relative binding determined? Why does only 25% of SDHAF2 or SDHAF4 bind to SDHA? Is it relative to input? A known amount of input should be included on the gel. When starting with the SDHA-SDHAF2 complex

(presumably at 1:1 stoichiometry) how is the further addition of SDHAF2 providing 30% relative binding? There should be no further binding as the complex is saturated.

All graphs on Figure S1 should include appropriate statistical information (N = ?). What do the error bars represent?

2. The authors have previously described the importance of dicarboxylate in the assembly (flavinylation) of apo-SDHA together with SDHAF2. Various dicarboxylates are present in mitochondria, however in these reductionist binding assays (shown in Figure 2) the authors have added FAD but not dicarboxylates (according to the method provided). The authors should repeat the experiments in Figure 2, in the presence and absence of dicarboxylate.

3. The position of molecular weight markers must be shown next to the BN-PAGE gel images to allow the readers to determine the apparent size of each of the complexes.

4. Migration of proteins to the same position on a native gel does not indicate proteins are part of the same complex. Page 4: Figure 2b, lane 1 alone does not show that SDHA comigrates with SDHAF2 as stated in the main text. A comparison of SDHA blot (Fig. 2b, Panel 1, lane 1) with the SDHAF2 blot (Fig. 2b, Panel 3, lane 1) suggests they migrate in a complex. The authors should revise the text to explain the data more precisely by directing the reader to the appropriate lanes and explaining the limitation of the data.

5. To help identify the composition of mitochondrial protein complexes observed following separation by native PAGE (Figure 2b) the team also analysed the migration of recombinant protein complexes of known composition i.e. SDHA alone, SDHA-SDHAF2, SDHA-SDHAF2-SDHAF4, SDHA-SDHAF4 and SDHAF4 alone. Unfortunately, this analysis was performed on separate gels and with respect to the recombinant protein complexes, was vastly overloaded forming broad bands and with no size markers provided to the side of the gel images. The team must at the very least run the recombinant protein complexes (ng quantities) alongside solubilised mitochondrial protein samples on the same native gel. Detection of the complexes can be via anti-SDHA immunoblotting just as performed in Figure 2b. The bands assigned in Figure 2b (e.g. SDHA-SDHAF4) should align with the band observed for recombinant SDHA-SDHAF4 complex. While not definitive, this analysis will provide stronger support for the authors' interpretation and model if indeed the complexes align as they suggest.

6. What is the evidence that the subcomplex in Figure 2b, Panel 1, lane 3 is a binary complex composed of SDHA and SDHAF4 as labelled on the side of the gel image? Is it based solely on the similar (but not the same) migration of a prominent band seen for the SDHAF4 blot (Figure 2b, Panel 4, lane 3)?

7. In Figure S2b the concentrations of SDHAF4 are shown alongside the gel image. Presumably these values represent the starting concentrations of the protein samples loaded on the gel. Were equal volumes of sample loaded onto both gels (native and SDS-PAGE)? What was the volume? It would be helpful to know the amount of protein loaded in each lane.

8. What is the assembly path of SDHA in the absence of SDHAF2? Can SDHA form a complex with SDHAF4 in mitochondria as seen via the in vitro binding assays with apo-SDHA (Figure 2a, lane 2)? This

could be simply analysed using the available SDHAF2 knock out cell line reported in the methods. On this point the authors mention on Page 4 that they knocked out SDHAF2 from the SDHAF4KO36 cell line and yet in the methods (page 11) they mention that the SDHAF4 knockout was generated in a verified SDHAF2 knock cell line. Which is it? Has the correct method been provided?

9. The following sentence appears on page 6 of the manuscript “Because the active site is synthesised in part by SDHAF2, this interaction helps to explain why SDHAF4 functions after SDHAF2” The authors should rephrase this sentence expanding their explanation regarding the type and order of interactions. For example, previously structural data published by this group highlighted the important interactions of SDHAF451 during flavinylation of SDHA, a residue that is solvent exposed in the SDHA-SDHAF2 complex. This same residue on SDHA makes direct contact with the conserved residues at the C-terminus of SDHAF4. Why? Is it important to prevent solvent exposure post-flavinylation of SDHA? Can SDHAF4 in fact act prior to flavinylation to regulate this event under certain cellular conditions? Again, it's surprising that the potential significance of this interaction has not been discussed.

10. Page 5. “Avidly” is used inappropriately. What do the authors really mean here? Furthermore, how do the authors reconcile the proposed activity of SDHAF4 in displacing SDHAF2 from SDHA when a stable recombinant SDHA-SDHAF2-SDHAF4 complex can be formed in vitro? Also, how does a dicarboxylate impact the structure and mechanism of assembly post flavinylation? Curiously the structure of the SDHA-SDHAF2-SDHAF4 complex does not contain a bound dicarboxylate however the SDHA-SDHAF4 complex contains bound malonate. The authors must point out this difference and explain whether it is significant with respect to the assembly pathway and structures observed. Could the SDHA-SDHAF4 complex be an off-pathway assembly intermediate? The authors own assembly schematic (Figure 6) shows dicarboxylate release from SDHA-SDHAF2 after the flavinylation step and prior to any interaction of the protein complex with SDHAF4.

11. There is little experimental evidence to suggest (see abstract) that the changes in intrinsic disorder of the assembly factors are required to release SDHA from metastable complexes. As more appropriately stated in the manuscript introduction, changes in intrinsic disorder of SDHAF2 and SDHAF4 may be important for release of SDHA. The abstract should be revised accordingly.

12. Was the recombinant SDHAF2 protein used in the previous study (Sharma et al., 2020) the same construct used in this study? This is important because the authors are claiming differences in the structural order of the N-terminus of SDHAF2 in complex with SDHA with and without SDHAF4 bound. It appears the N-terminus of SDHAF2 used in the above-mentioned study retained part (seven amino acid residues) of the mitochondrial targeting signal (which is normally removed in mitochondria) whereas the construct used in this study did not. The authors need to highlight this difference and comment on whether it could impact the structures observed.

13. Likewise, the wwPDB X-ray structure validation reports provided alongside this manuscript for reviewing purposes show the full sequence of the protein constructs successfully used to grow crystals for structural analysis. Alarmingly this reveals the human SDHAF4 recombinant protein present in the ternary SDHA-SDHAF2-SDHAF4 complex was the precursor protein i.e. it contains the entire mitochondrial targeting sequence which in mitochondria is cleaved by processing peptidases. This same

processing does not occur in the *E. coli* expression host when produced recombinantly. As such, this protein construct is quite artificial and certainly not entirely physiologically relevant. Surprisingly, a different recombinant form of SDHAF4 was used to form crystals of the binary complex. This construct contains the mature SDHAF4 bearing an N-terminal His-TEV. The use of two different protein constructs cast doubt on the significance of the reported changes in the structural order of the N-terminal region of SDHAF4 observed between the two structures. Are the structural differences observed by the authors simply a consequence of incorrectly using the precursor protein instead of the mature protein. Without the validation reports such detail cannot be gleaned from the manuscript due to the lack of detail provided regarding the proteins used. The authors must show (in the supplementary section) the primary sequence of all recombinant protein constructs used in the biochemical and structural studies. They should unambiguously state, which protein constructs were used in which experiments.

14. In regard to the trimeric complex, the authors appear to be having a bet each way. In a previous publication in *Nature Communications* (Bezawork-Geleta et al., 2018 - PMID: 29880867), the group explicitly stated that SDHAF2 and SDHAF4 together with SDHA constituted the 100 kDa complex (which they termed CII-low), but now the authors suggest that the underlying molecular species was unclear (see page 7 of the discussion), and despite presenting the structure of the trimeric complex, the authors appear to shy away from the idea this complex appears on native gels in mitochondria. The authors should acknowledge that the previous data was likely incorrectly interpreted, and the data presented here suggests the 100 kDa complex is made up of only SDHA and SDHAF2.

15. Page 10: A reference for the preparation of apo-SDHA should be provided.

Page 10: Binding assays contained 25 μ M of imidazole? I assume this is a typographical error, should it be 25 mM imidazole?

Page 10: "... on any kDa gels..." What does this mean?

Page 10; ...stained with InstantBlue Coomassie gel" Should probably be "....stained with InstantBlue Coomassie stain...."

Reviewer #2 (Remarks to the Author):

This manuscript by Sharma and colleagues defines, at the molecular level, the first steps of maturation and assembly of the mitochondrial respiratory chain complex II and TCA cycle enzyme succinate dehydrogenase. The complex is formed by four subunits and the manuscript focuses on the maturation and assembly of subunit SDHA, which contains an FAD co-factor. Three intermediates are identified here, containing SDHA and the assembly factor SDHAF2, SDHA-SDHAF2 and the assembly factor SDHAF4, and SDHA-SDHAF4. The binding of the assembly factors to SDHA involves a change in disorder, which also facilitates the displacement of AF2 by AF4. In cellulo, and in vitro studies are well combined to demonstrate the order of events. The discussion is rich in finding parallelisms with other assembly

systems, in which changes in disorder in assembly factors appear to be a common theme. Therefore, the conclusions have a broader impact in multiple biological systems. The manuscript is technically and conceptually sound and in the opinion of this reviewer should be published as it is.

Reviewer #3 (Remarks to the Author):

In their manuscript Sharma et al. describe how one of the subunits, SDHA in the mitochondrial CII complex is matured. The CII complex is essential for cell respiration and metabolism, and its misassembly is associated with several diseases, including ischemia and cancers. The authors have used pull-down assays, western blots and biophysical methods, including structure determination by x-ray crystallography to elucidate the order of the events in maturation of SDHA.

Although the manuscript is well written and the experimental work is solid, some clarity in the take home message of the work is missing. I am thus not sure if the authors main conclusion is the sequence of maturation events presented in Figure 6, which I find needs further justification (*vide infra*) - or if it is how disorder in SHDAF2 and SHDAF4 is important for this and other assembly processes as discussed on pages 8 - 9 in the manuscript. It thus also becomes unclear exactly how the authors see their results advance the field.

In addition to the overall criticism, I have the following more specific points.

1. On the cover page it is confusing that the addresses are not listed in numerical order.
2. p2, last sentence in the abstract. Is this the most important insight from the work and where the authors see its largest impact? What about the biological insight that understanding the assembly process give?? This is not mentioned here and is also largely missing from the discussion section of the manuscript.
3. p3, first paragraph of the results section. This is mainly background information and should be put in the introduction.
4. p4, first paragraph and Figure 2a. Is only SDHA loaded with FAD tested in the pull-downs? How will the interaction pattern change if no FAD is present, if FAD is covalently bound and if dicarboxylate is added? Although some of this is mentioned in the previous paragraph it would be great to have these effects shown in the same pull-down assay where the other interactions are studied. It is also very hard to read the gel in Figure 2A as no labelling of the lanes are put in the figure.
5. p4, first paragraph. The authors should specifically mention which previous results in the literature that supports that SDHAF2 is the first factor to bind. That SDHAF2 has a lower K_d than SDHAF4 as the pull-down assay suggests does not necessarily mean that SDHAF2 binds first. E.g. the cellular concentrations could favour SDHAF4.

6. p4, last paragraph. From the native gels from the KO cells the authors conclude that SDHA-AF4 is not forming in cells. How is this consistent with the pull-down assays where both AF2 and AF4 form complexes with SDHA?

7. p5, last paragraph. The authors mention CD spectra, which they also show in Fig S4. How these experiments were made is not written in the methods section. The CD spectrum in Figure S4b will be dominated by the folded SDHA that gives a strong signal. A spectrum of apo-SDHA should be subtracted to assess the structure of SDHAF4 in the complex!

8. p5, last paragraph. The authors write that they have made a chemical shift analysis. I can't find where this is reported. It would be highly interesting to see if there is any low populated transient structure. Also in the ¹⁵N HSQC, there is a set of low intensity peaks suggesting that SDHAF4 is also found in an alternative conformation - and thus not fully disordered? Or is this an impurity? Showing SEC or SDS-PAGE of the sample would be appropriate.

9. p6, section "SDHA-AF4 is the Third Protein-Protein Complex in SDHA Maturation" The SDHA:SDHAF4 complex for the crystal structure is prepared by adding excess SDHAF4 relative to SDHAF2. However, in the model presented in Figure 6 there is no concentration dependence going from the third to fourth state. So what is the driving force for releasing SDHAF2 from SDHA? The most obvious suggestion is that SDHAF2 and SDHAF4 interact with each other and that excess of either of these components will drive the other component out of the complex with SDHA, by forming a SDHAF2:SDHAF4 complex. This should be tested.

10. p7, first line. The authors write "To test whether *intrinsic disorder* is important for the transfer...". From their experiments what the authors are testing is whether the disordered regions (not disorder in itself) are important for the interaction. They are not testing if any disordered sequence is working or if it is possible to replace the disordered regions with ordered regions. At the end of the second paragraph on p7 the authors correctly conclude that "These findings implicate the *intrinsically disordered regions* in the process..."

11. p7, second paragraph of the discussion. The authors write "We show that SDHA proceeds through at least three metastable intermediates (Figure 6) prior to being incorporated into CII..." The authors have not shown the incorporation of SDHA into CII.

12. Figure 2. Labels are missing on the lanes of the gel in panel a. The figure legend for panel b report that the gels are representative of three experiments. In Fig S1a that shows the quantification there are 4-5 data points on each bar. The same goes for panel b, where there are 3-4 points in Fig S1b-h.

13. Figure 3. It is almost impossible to see the FAD in panel a, although the legend to panel a describes how FAD is displaced. In panel d, the labels on the lanes of the gel are missing.

14. The authors should carefully proofread the manuscript, as there are several missing words throughout the text.

REVIEWER COMMENTS

Reviewer #1 (Remarks to the Author):

In this manuscript "Disordered-to-ordered transitions in assembly factors allow complex II catalytic subunit to switch binding partners" by Sharma et al., the authors present the first atomic resolution structure of the human complex II assembly factor SDHAF4 in a ternary complex with the flavo-protein holo-SDHA and the assembly factor SDHAF2 and additionally in a binary complex with holo-SDHA. The ternary complex presented here shows that SDHAF2 and SDHAF4 occupy distinct binding sites on SDHA. Interestingly the conserved C-terminal residues in SDHAF4 interact directly with amino acid residues within the active site of holo-SDHA. The structures presented here (when compared to the published human SDHA-SDHAF2 complex and yeast SDHAF2) illustrate changes in intrinsic disorder of the assembly factors in the free and complexed states. In the abstract the authors state that a gain of structural order is required for release of SDHA from its assembly factors.

The team also provided data from in vitro binding studies and native gel analysis of recombinant protein complexes and an analysis of steady state protein complexes in mitochondria isolated from parental and selected knockout cell lines. They have attempted to draw conclusions from the data about the order of assembly of complex II and the composition of intermediates by comparing relative migration on separately run native gels (see further specific comments below in point 5). The analysis of the SDHAF4 knock out cell line does show clear accumulation of a low molecular weight complex (which based on relative migration is likely, but not definitively, the SDHA-SDHAF2 complex) suggesting SDHAF4 is required for progression of SDHA along the assembly pathway. The in vitro binding studies support such a role to some degree, however these data are not incontrovertible. Furthermore, the statement in the abstract that SDHAF4 must displace SDHAF2 for assembly to proceed is poorly supported by the data presented. A significant amount of fully assembled SDHA is seen in the SDHAF4 knock out cell line (Figure 2b, panel 1, lane 2). Rather, assembly of SDHA into the terminal complex is more efficient when SDHAF4 is present in mitochondria but SDHAF4 is not essential. The rate of dissociation of SDHAF2 from SDHA is potentially enhanced via the action of SDHAF4.

The following points must be addressed by the authors.

1. Figure 2a is not labelled sufficiently. It is unclear which experiment is shown in which lane due to the lack of information provided in the figure legend, main text or through labelling. The information can only be assumed from the key and figure legend on Figure S1. The authors should provide appropriate labelling for Figure 2a so it can be critically analysed. The authors also provide a quantitation of Figure 2a on Figure S1, however here the figure shows error bars, although the figure legend implies it is from the quantitation of a single gel (Figure 2a). Does this quantitation represent technical replicates of a single experiment ($N = 1$), or alternatively from 4 or 5 individual experiments ($N = 4$ or 5) as suggested by the number of points on the graph. Related to the quantitation on Figure S1a, how was the relative binding determined? Why does only 25% of SDHAF2 or SDHAF4 bind to SDHA? Is it relative to input? A known amount of input should be included on the gel. When starting with the SDHA-SDHAF2 complex (presumably at 1:1 stoichiometry) how is the further addition of SDHAF2 providing 30% relative binding? There should be no further binding as the complex is saturated.

All graphs on Figure S1 should include appropriate statistical information ($N = ?$). What do the error bars represent?

We appreciate both reviewers identifying ways to improve the in vitro assay. In response to comments from both R1 and R3 on the content and clarity of Figure 2a, we fully redesigned this experiment to make it more systematic. The revised experiment allows the reader to step through the SDHA maturation process more logically and better explains the status of SDHA induced by each condition. As described in more detail below, we also pre-saturated the SDHA-AF2 complex prior to initiating the pull-down in order to eliminate confusion resulting from a heterogeneous starting sample. Per the reviewer's request, this revised experiment includes dicarboxylate as a variable and shows input. The associated display item now contains five panels and more thorough labeling. We clarified in the legend that the gel shown in the figure was representative and that four replicates were performed with protein produced from independent preparations. Finally, due to expanding the experiment, the panel containing the knockout cell studies (formerly Fig. 2b) has been moved to a separate figure, and all figures in the manuscript have been renumbered.

In addition, the reviewer was curious why the addition of SDHAF2 to the SDHA-AF2 sample showed an increase in the amount of SDHAF2 bound. In the original experiment, the starting material came from SDHA and SDHAF2 coexpressed in *E. coli* and purified by an affinity tag on SDHA. There is no a priori reason why the binding would have been saturated under these conditions. Thus, the measured binding of SDHAF2 likely reflected that the SDHA-AF2 sample, as prepared in the original manuscript, was not saturated and was a mixture of SDHA and SDHA-AF2. In the redesigned experiment, we pre-saturated the SDHA with SDHAF2 prior to initiating the assay, and the increase in

SDHAF2 binding is therefore no longer observed. We thank the reviewer for pointing this out as it helped to give a clearer result for readers.

2. The authors have previously described the importance of dicarboxylate in the assembly (flavinylation) of apo-SDHA together with SDHAF2. Various dicarboxylates are present in mitochondria, however in these reductionist binding assays (shown in Figure 2) the authors have added FAD but not dicarboxylates (according to the method provided). The authors should repeat the experiments in Figure 2, in the presence and absence of dicarboxylate.

As described in point 1, we repeated this experiment more systematically, under a broader range of conditions and with additional controls, including in the presence of dicarboxylate.

3. The position of molecular weight markers must be shown next to the BN-PAGE gel images to allow the readers to determine the apparent size of each of the complexes.

We thank the reviewer for pointing this out. Molecular weight markers are now shown either as their positions or as the original marker.

4. Migration of proteins to the same position on a native gel does not indicate proteins are part of the same complex. Page 4: Figure 2b, lane 1 alone does not show that SDHA comigrates with SDHAF2 as stated in the main text. A comparison of SDHA blot (Fig. 2b, Panel 1, lane 1) with the SDHAF2 blot (Fig. 2b, Panel 3, lane 1) suggests they migrate in a complex. The authors should revise the text to explain the data more precisely by directing the reader to the appropriate lanes and explaining the limitation of the data.

We agree with the reviewer that comigration does not indicate that proteins are bound in the same complex. As suggested by the reviewer, we performed additional analysis (below) to help assign complexes and modified the text to better explain which of the proposed complexes are consistent with the observed bands and which are yet to be fully explained.

To address this more fully, we performed experiments that more strongly support assignment of the various complexes: immunoprecipitation and complexome analysis. The immunoprecipitation identifies that SDHAF4 only binds robustly to SDHA in cells with the ~150 kD band. This is now included as a panel in Figure 3. Because migration of the band observed in cells differs from the migration of the purified ASDHA-AF4 complex, we agree with the reviewer that we can only say that SDHA and SDHAF4 are in the same complex (and not that this is the SDHA-AF4 complex). The major complex that accumulates in cells likely contains additional components. We do observe a band in the cells consistent with the SDHA-AF4 complex, which is likely on-pathway to this higher molecular weight intermediate.

The complexome analysis (Figure R1, below) was not of publication quality but also supports the assignment of the ~100 kD band as SDHA-AF2, the ~120 kD band as SDHA-AF4, and the ~150 kDa band as containing both SDHA-AF4.

Figure R1. Complexome analysis of the accumulated species in knockout cell lines. (Left) representative Western analysis of the accumulated species from the indicated cell lines. (Right) Mass spectrometry of the gels shows that the ~100 kD band contains SDHA and SDHAF2.

5. To help identify the composition of mitochondrial protein complexes observed following separation by native PAGE (Figure 2b) the team also analysed the migration of recombinant protein complexes of known composition i.e. SDHA alone, SDHA-SDHAF2, SDHA-SDHAF2-SDHAF4, SDHA-SDHAF4 and SDHAF4 alone. Unfortunately, this analysis was performed on separate gels and with respect to the recombinant protein complexes, was vastly overloaded forming broad bands and with no size markers provided to the side of the gel images. The team must at the very least run the recombinant protein complexes (ng quantities) alongside solubilised mitochondrial protein samples on the same native gel. Detection of the complexes can be via anti-SDHA immunoblotting just as performed in Figure 2b. The bands assigned in Figure 2b (e.g. SDHA-SDHAF4) should align with the band observed for recombinant SDHA-SDHAF4 complex. While not definitive, this analysis will provide stronger support for the authors' interpretation and model if indeed the complexes align as they suggest.

We thank the reviewer for pointing this out. In response, we separated lysates from SDHAF2^{KO}/SDHAF4^{KO}, each with SDHAF2 and SDHAF4 overexpressed on a plasmid. We separated these on a 4 – 16% native gel along with purified protein used for crystallization that contained SDHA, SDHAF2, and SDHAF4. We then performed Western analysis (Figure R2, left). We note that the migration is consistent with the migration shown in the main text (also on 4 – 16% gradient gels). However, because native gels are prone to anomalous migration, these appear to differ slightly in terms of exact molecular weight when compared to samples separated on single % native gels, as in Figure S3 (a 10% native gel). Nevertheless, the conclusions made from samples separated on two distinct types of gels are consistent with each other.

Specifically, this analysis shows four distinct bands across the lanes in these gels. On the 4 – 16% gradient gels, these bands are centered at ~100 kDa, ~120 kDa, ~150 kDa, and ~200 kDa.

The ~100 kDa band appears in both samples from cells and the purified protein and stains for SDHA and SDHAF2, further supporting our assignment of this band as the SDHA-AF2 complex, as in the original manuscript. The ~120 kDa band is observed in both cells and the purified protein and contains SDHA and SDHAF4, strongly supporting its assignment as the SDHA-AF4 complex. The ~150 kDa band is only observed in the cells and contains SDHA and SDHAF4, supporting this as a complex containing SDHA, SDHAF4, and another species that is yet to be determined. Finally, the ~200 kDa band, which is assigned as assembled CII, only appears in cells (as expected).

The most probable explanation for this result is that the complex in cells contains another protein that has not been identified. This could suggest that the SDHA-AF4 complex in cells quickly binds to another assembly chaperone. We are in the process of identifying whether this is a known protein (for example, SDHAF1 or SDHAF3) versus a novel assembly chaperone. Because the identification of a distinct assembly complex is outside of the focus of this manuscript, its inclusion might dilute the take-home message and we have therefore not included that result in the revision. Instead, we modified the language surrounding the assignment of the SDHA-AF4 complex to a specific band. The data continue to very strongly support the ~100 kDa band as SDHA-AF2 and all evidence shows that SDHAF4 can displace SDHAF2 from SDHA in vitro, indicating this as an intermediate step in assembly.

Figure R2 (left). Migration of accumulated species in cells as compared to purified proteins.

6. What is the evidence that the subcomplex in Figure 2b, Panel 1, lane 3 is a binary complex composed of SDHA and SDHAF4 as labelled on the side of the gel image? Is it based solely on the similar (but not the same) migration of a prominent band seen for the SDHAF4 blot (Figure 2b, Panel 4, lane 3)?

We agree with the reviewer that comigration does not indicate that proteins are bound in the same complex. As described above, we additionally performed immunoprecipitation and complexome analysis to support the assignment of these bands. As suggested by the reviewer, we modified the text and figure labeling to indicate that this complex contains SDHA and SDHAF4.

7. In Figure S2b the concentrations of SDHAF4 are shown alongside the gel image. Presumably these values represent the starting concentrations of the protein samples loaded on the gel. Were equal volumes of sample loaded onto both gels (native and SDS-PAGE)? What was the volume? It would be helpful to know the amount of protein loaded in each lane.

We thank the reviewer for their close attention to detail. Yes, equal volumes were loaded in this concentration series. To clarify this, we modified the figure labels and figure legend to indicate the quantity of protein that was run on the gel rather than the concentration of the sample, now Figure S3b.

8. What is the assembly path of SDHA in the absence of SDHAF2? Can SDHA form a complex with SDHAF4 in mitochondria as seen via the in vitro binding assays with apo-SDHA (Figure 2a, lane 2)? This could be simply analysed using the available SDHAF2 knock out cell line reported in the methods.

We thank the reviewer for this thoughtful comment, where the true response is relatively deep in nature. In fact, it is an area of active research. With the reviewer's grace, a full explanation of why we observe this association in vitro requires some initial and long explanation of preferred versus non-preferred protein maturation pathways.

There are likely both preferred and non-preferred pathways of maturation in cells for Complex II (as well as any protein in the cell). Non-preferred pathways of maturation include (but are not limited to) SDHAF2-independent FAD attachment or SDHAF4 binding to unflavinylated SDHA. These and other non-preferred paths of maturation are unlikely to be fully excluded but may not be observable in cells when the preferred pathway is functioning correctly, i.e. in the presence of functional SDHAF2. These non-preferred pathways and interactions might only appear when concentrations of some components are greatly increased. This could occur in the in vitro assays with high protein concentrations or could occur when cells are perturbed, as would be observed in a patient following a heart attack or stroke or with a point mutation in SDHAF2 that affects SDHA binding.

These less-preferred pathways of protein maturation are observed in the in vitro work (Figure 2b), and the perturbed cells (Figure 3b) and can be predicted by using a chemical approach and mass action analysis. For the SDHA subunit of complex II, this mass-action diagram is relatively complex (see Figure R3, below). The presence of non-preferred pathways of maturation during SDHA maturation (or the maturation of any protein) explains a number of findings in the literature that seemed to contradict. For example, minor amounts of Complex II can assemble when SDHAF4 is knocked out in model organisms. In addition, SDHAF2 enhances covalent flavin attachment, but some cancer cells are able to assemble Complex II with covalent FAD following the knockout of SDHAF2.

While a full discussion of alternative assembly mechanisms under disease conditions is outside of the scope of this work, we are actively collecting data toward better defining this. These data use a computational approach in conjunction with the mass action diagram with a goal of developing a predictive and quantitative model for Complex II assembly under normal versus perturbed cellular conditions. Some of the preliminary data for this next manuscript are relevant to the reviewer's query. For example, to identify which non-preferred pathways of Complex II maturation can be accessed under highly perturbed conditions, we overexpressed SDHAF4 in the SDHAF2 KO cell line, as requested (now included in the revised Figure 3). We are in the process of identifying what conditions would promote that interaction in cells. As another part of this future manuscript, we show that SDHAF4 knockout appears to result in a lower percentage of SDHA containing covalent flavin, even though SDHAF4 is not involved in covalent flavinylation either directly or indirectly. Instead, each of these results suggests that SDHA maturation is proceeding via a non-preferred pathway when cellular conditions are sufficiently perturbed and that this results in less-preferred protein associations. These types of situations might occur physiologically in a patient with a homozygous knockout of SDHAF2 or SDHAF4. We believe that we captured this non-preferred association in vitro by using concentrations that drove the association equilibrium toward a less-favored complex.

Given the focus of this manuscript, we did not add a deep delve into preferred versus non-preferred pathways to the discussion. Instead, we plan to complete a future manuscript that rigorously tests how Complex II can mature under non-preferred conditions during disease. Some of the data in this response to review will be included in that manuscript.

Instead, to improve the manuscript, we modified the assay shown in Figure 2a. First, we ran additional controls that lay out the experiments more logically. In addition, we recognized that the mass action diagram predicts a concentration-dependence of SDHAF4 bindings to SDHA, i.e. what we would predict to be a non-preferred but non-excluded interaction in cells. Based on this, we adjusted the concentration levels of the assay such that a lower level of SDHAF4 binds (still not zero) under the conditions tested, thus sampling the non-preferred pathway less frequently. The revised manuscript makes it clear that we do observe SDHA binding to SDHAF4 at a non-zero level when FAD is present.

Figure R3. Mass action diagram detailing all possible assembly paths for SDHA in a system of binary interactions with four binding partners (FAD, dicarboxylate, SDHAF2, and SDHAF4). We believe that the most favorable route of biological maturation is shown in the present manuscript. However, alternative handling may occur under conditions with perturbed cellular conditions, including in knockout cells, with disease-associated mutations, or during metabolic conditions (such as hypoxia) with accumulated dicarboxylates, particularly fumarate. Some of these possible perturbations are now represented by the augmented studies shown in the revised Figure 3.

On this point the authors mention on Page 4 that they knocked out SDHAF2 from the SDHAF4KO36 cell line and yet in the methods (page 11) they mention that the SDHAF4 knockout was generated in a verified SDHAF2 knock cell line. Which is it? Has the correct method been provided?

We appreciate the reviewer's attention to detail. The double knockout was made by deleting SDHAF2 from the SDHAF4-36KO line. We corrected this in the methods.

9. The following sentence appears on page 6 of the manuscript "Because the active site is synthesised in part by SDHAF2, this interaction helps to explain why SDHAF4 functions after SDHAF2" The authors should rephrase this sentence expanding their explanation regarding the type and order of interactions. For example, previously structural data published by this group highlighted the important interactions of SDHAR451 during flavinylation of SDHA, a residue that is solvent exposed in the SDHA-SDHAF2 complex. This same residue on SDHA makes direct contact with the conserved residues at the C-terminus of SDHAF4. Why? Is it important to prevent solvent exposure post-flavinylation of SDHA? Can SDHAF4 in fact act prior to flavinylation to regulate this event under certain cellular conditions? Again, it's surprising that the potential significance of this interaction has not been discussed.

We appreciate reviewer's attention to detail regarding the sentence "Because the active site is synthesized in part by SDHAF2, this interaction helps to explain why SDHAF4 functions after SDHAF2". In the revised version we removed that sentence and instead added more detail on the interactions between the SDHA active site residues with C-

terminus SDHAF4 residues, explaining how this may be important for eliminating off-pathway catalysis in the assembly intermediate.

Specifically, the penultimate SDHAF4 C-terminal residue SDHAF4^{D107} makes strong interactions with the active site residues of SDHA^{H407} and SDHA^{R451}. The latter residues are also mutated in patients with CII insufficiency. Moreover, the C-terminal SDHAF4^{F108} occludes the location that would be occupied by the dicarboxylate substrate in assembled CII. Taken together, the two terminal residues of SDHAF4, i.e. SDHAF4^{D107} and SDHAF4^{D108}, likely preclude the binding of substrates or inhibitors to the SDHA subunit during the CII assembly process. They may also support the unusually strong binding of SDHAF4 to SDHA, and the ability to displace the tightly-bound SDHAF2.

The modified studies in Figure 2 suggest that SDHAF4 is unlikely to bind SDHA prior to flavinylation or formation of the SDHA-SDHAF2 assembly intermediate in the (healthy) cellular environment. However, if such a complex did form, we anticipate that it will be a non-productive, off-pathway assembly intermediate, as suggested by our preliminary data for the next manuscript.

10. Page 5. "Avidly" is used inappropriately. What do the authors really mean here? Furthermore, how do the authors reconcile the proposed activity of SDHAF4 in displacing SDHAF2 from SDHA when a stable recombinant SDHA-SDHAF2-SDHAF4 complex can be formed in vitro? Also, how does a dicarboxylate impact the structure and mechanism of assembly post flavinylation? Curiously the structure of the SDHA-SDHAF2-SDHAF4 complex does not contain a bound dicarboxylate however the SDHA-SDHAF4 complex contains bound malonate. The authors must point out this difference and explain whether it is significant with respect to the assembly pathway and structures observed. Could the SDHA-SDHAF4 complex be an off-pathway assembly intermediate? The authors own assembly schematic (Figure 6) shows dicarboxylate release from SDHA-SDHAF2 after the flavinylation step and prior to any interaction of the protein complex with SDHAF4.

At the reviewer's request, we removed the word 'avidly' from the revised manuscript.

In terms of reconciling the role of SDHAF4 in removing SDHAF2 from SDHA with the observation that we could stabilize intermediate SDHA-AF2-AF4 structure, we note that CII assembly normally occurs at 37 °C in human cells. In order to capture the SDHA-AF2-AF4 intermediate, we prepared the sample at 4 °C to slow down SDHAF2 release. Raising the temperature of the SDHA-AF2-AF4 to room temperature overnight or 37 °C for 4-5 hr allowed for the spontaneous release of SDHAF2 from this intermediate. Together, we believe that this indicates that the proteins are functional but that the exchange of SDHAF2 for SDHAF4 was slowed by lowering the temperature. This is described in the methods. To clarify this for readers, we have now augmented the results with an overview of the preparation information and a rationale for why this preparation would have captured a normally-transient intermediate.

We thank the reviewer for pointing out the dicarboxylate, which may be of interest to specialists in the field. The short answer is that there are more than one non-equivalent dicarboxylate binding sites in SDHA that are functionally relevant. The text may have unintentionally conflated these as the reviewer points out changes in dicarboxylate occupancy at distinct sites.

The longer answer is that there is one distinct dicarboxylate binding site at the active site FAD, however, dicarboxylate binds at this site in distinct poses when assembled CII is compared to the SDHA-AF2 structure. The first dicarboxylate pose correlates with succinate \leftrightarrow fumarate interconversion in assembled CII and is likely the site of on-pathway substrate binding during catalysis in the Krebs cycle. The second, distinct, binding pose for dicarboxylate may enhance covalent flavinylation, with fumarate preferred (<https://pubmed.ncbi.nlm.nih.gov/32887801/> and <https://pubmed.ncbi.nlm.nih.gov/36089066/>).

Dicarboxylate cannot bind at the FAD at the same time as SDHAF4, because SDHAF4 occludes the binding with its C-terminus (Figure 4c). This active site dicarboxylate is therefore shown leaving prior to SDHAF4 binding in the second step of Fig. 6.

As the reviewer points out, malonate is associated with the structure of the SDHA-AF4 complex in the PDB validation report. This binding position for malonate is distal from the active site and does not have a known role in flavinylation or catalysis. The studies here do not inform whether there is a functional role of this malonate, or whether this was simply a crystallization artifact because malonate was in the crystallization buffer. We may investigate this in the future.

To clarify the role of different dicarboxylate binding sites during CII maturation for the reader, we expanded both the results section and the legend to Figure 7 (original Figure 6). The revised text now explicitly indicates where and when dicarboxylate is known to act during SDHA maturation. It also more clearly states that the canonical dicarboxylate site

at FAD is occluded when SDHAF4 binds. In the revised legend, we explicitly indicate that there is a malonate bound in the SDHA-AF4 structure at a distal site, but that this second binding site is of unknown relevance.

11. There is little experimental evidence to suggest (see abstract) that the changes in intrinsic disorder of the assembly factors are required to release SDHA from metastable complexes. As more appropriately stated in the manuscript introduction, changes in intrinsic disorder of SDHAF2 and SDHAF4 may be important for release of SDHA. The abstract should be revised accordingly.

We thank the reviewer for opening this discussion. In response to this query and comments from reviewer 3, we substantially modified the abstract and sections of the text that refer to the intrinsic disorder. The modified version is clearer that this tests the regions of the protein that are intrinsically disordered.

12. Was the recombinant SDHAF2 protein used in the previous study (Sharma et al., 2020) the same construct used in this study? This is important because the authors are claiming differences in the structural order of the N-terminus of SDHAF2 in complex with SDHA with and without SDHAF4 bound. It appears the N-terminus of SDHAF2 used in the above-mentioned study retained part (seven amino acid residues) of the mitochondrial targeting signal (which is normally removed in mitochondria) whereas the construct used in this study did not. The authors need to highlight this difference and comment on whether it could impact the structures observed.

We thank the reviewer for their close attention to detail. We explicitly confirm that we used the same construct for SDHAF2 as was used in Sharma et al (2020) *PNAS*. We further confirm that the plasmid is as stated and referenced in the first sentence of the methods section (plasmids section) and in the section for protein purification for crystallography. We modified the methods to ensure that this is clearer.

13. Likewise, the wwPDB X-ray structure validation reports provided alongside this manuscript for reviewing purposes show the full sequence of the protein constructs successfully used to grow crystals for structural analysis. Alarmingly this reveals the human SDHAF4 recombinant protein present in the ternary SDHA-SDHAF2-SDHAF4 complex was the precursor protein i.e. it contains the entire mitochondrial targeting sequence which in mitochondria is cleaved by processing peptidases. This same processing does not occur in the *E. coli* expression host when produced recombinantly. As such, this protein construct is quite artificial and certainly not entirely physiologically relevant. Surprisingly, a different recombinant form of SDHAF4 was used to form crystals of the binary complex. This construct contains the mature SDHAF4 bearing an N-terminal His-TEV. The use of two different protein constructs cast doubt on the significance of the reported changes in the structural order of the N-terminal region of SDHAF4 observed between the two structures. Are the structural differences observed by the authors simply a consequence of incorrectly using the precursor protein instead of the mature protein. Without the validation reports such detail cannot be gleaned from the manuscript due to the lack of detail provided regarding the proteins used. The authors must show (in the supplementary section) the primary sequence of all recombinant protein constructs used in the biochemical and structural studies. They should unambiguously state, which protein constructs were used in which experiments.

We thank the reviewer for their close attention to detail. We explicitly confirm that the construct used for the expression of SDHAF4 in *E. coli* lacked the signal sequence and that all start sites are explicitly indicated in the methods.

We believe that the reviewer's concerns may stem from the PDB deposition process, which requires that both the native, full-length sequence as well as the sequence variations that appeared in the construct. In this case, it appears that we unintentionally swapped these two sequences when uploading the SDHA-AF4 structure. We modified the PDB entry and provide an updated validation report. For the SDHA-AF2-AF4 structure, page 10 of the validation report gives the sequences used in the actual constructs for SDHA-AF2-AF4 structure, including tags and linkers. For the SDHA-AF4 validation report, page 9 shows the sequences. These sequences are both as described in the methods, where start sites are included in the explicit descriptions of each construct.

To further improve clarity, we added a sentence to the methods that all constructs developed for protein expression in *E. coli* lacked the mitochondrial targeting sequence.

14. In regard to the trimeric complex, the authors appear to be having a bet each way. In a previous publication in Nature Communications (Bezawork-Geleta et al., 2018 - PMID: 29880867), the group explicitly stated that SDHAF2 and SDHAF4 together with SDHA constituted the 100 kDa complex (which they termed CII-low), but now the authors suggest that the underlying molecular species was unclear (see page 7 of the discussion), and despite presenting the

structure of the trimeric complex, the authors appear to shy away from the idea this complex appears on native gels in mitochondria. The authors should acknowledge that the previous data was likely incorrectly interpreted, and the data presented here suggests the 100 kDa complex is made up of only SDHA and SDHAF2.

We thank the reviewer for opening this discussion but disagree with their interpretation of the Bezawork-Geleta paper. We further disagree with the reviewer's interpretation of our underlying rationale for modifying our hypothesis in the face of new data as 'having a bet each way'.

The ~100 kD complex containing SDHA been reported by multiple groups but there has not previously been a consensus on the molecular species that form this band. The first observation was over a decade ago by the Grimm group (see <https://www.nature.com/articles/cdd201093>). At that time, the band was only assessed for the presence of SDHA. Because of the molecular weight and the presence of SDHA, this was originally assigned as an SDHA-SDHB complex. It was not until 2018 (see <https://www.ncbi.nlm.nih.gov/pmc/articles/PMC5992162/>) that SDHB was shown to be absent from this complex. Here Bezawork-Geleta used affinity capture and mass spectrometry to interrogate this band, stating "The analysis revealed the presence of the CII assembly factors SDHAF2 (16.7 kDa) and SDHAF4 (9.9 kDa)." This wording was very intentional because the analysis did not unambiguously define the molecular nature of the complex. Indeed, a caveat of mass spectrometry is that species at low abundance can be shown to be 'present'. As a result, there was great care in ensuring that the wording did not overstep the limits of the technique. Most recently (and subsequent to the initial submission but now included in the revision), macrophage activation was shown to be associated with CII disassembly and the appearance of a 100 kD species containing SDHA but not SDHB (<https://www.science.org/doi/full/10.1126/sciadv.ade8701>). This band was interpreted as "SDHA monomer or SDHA bound to other proteins such as assembly factors." but was not probed for the presence of those assembly factors. All in all, none of this past literature definitively identifies the molecular assembly that forms the ~100 kD SDHA-containing band.

Here, we use purified proteins to show that SDHA-AF2, SDHA-AF2-AF4, and SDHA-AF4 migrate distinctly. We go on to show that isolated SDHA-AF2 migrates at ~100 kD while isolated SDHAF4 oligomerizes and that one of its oligomers coincidentally migrates at ~100 kD. We, therefore, believe that the most likely interpretation is that the ~100 kDa band that is induced during ischemia, acidification, cancer metabolism, and inflammation is due to an increase in the SDHA-AF2 complex. We further posit that some SDHAF4 was detected in a mass spectrometry experiment for multiple reasons. Isolated SDHAF4 oligomers can coincidentally migrate at the same molecular weight. In addition, SDHA-AF2-AF4 and SDHA-AF4, while centered at a distinct molecular weight, each run as broad bands that overlap with the ~100 kDa band. An increase in higher molecular weight bands is not induced under pathological conditions, suggesting that SDHAF4 that is migrating with these complexes may, again, coincidentally be picked up in an analysis of the components migrating at 100 kDa. Finally, we note that the limitations of mass spectrometry prevent definitive assignment. Mass spectrometry is a sensitive and qualitative technique that can simply identify the species present, regardless of abundance.

Taken together, we believe that our wording does not imply a 'bet each way' but instead is the fairest and most reasonable interpretation of the data in hand, including the appropriate consideration of caveats of the different techniques that were used.

Given the reviewer's concern, we modified both the introduction and the discussion to improve the language precision. This now indicates that the SDHA-AF2 is likely the species that increases the most under pathological conditions, but that we cannot exclude other species increasing in abundance as well. We have now included a citation for a 2023 study, published after our initial submission, showing that SDHA is released from CII and instead is found as a ~100 kD species during macrophage activation.

15. Page 10: A reference for the preparation of apo-SDHA should be provided.

Page 10: Binding assays contained 25 μ M of imidazole? I assume this is a typographical error, should it be 25 mM imidazole?

We appreciate the reviewer's attention to detail. We now included the reference for the preparation of apo-SDHA and confirm the 25 μ M imidazole concentration.

Page 10: "... on any kDa gels..." What does this mean?

"Any kD gels" is a product from BioRad (see the link to the manufacturer's site, below). We agree with the reviewer that the name of this product can cause unintentional confusion in the context of a sentence. To help improve this, we capitalized "Any", placed quote marks around the term "Any kD", and modified the sentence to be "...on "Any kD™" pre-cast protein gels..."

<https://www.bio-rad.com/en-us/sku/4569033-any-kd-mini-protean-tgx-precaster-protein-gels-10-well-30-ul?ID=4569033>

Page 10; ...stained with InstantBlue Coomassie gel" Should probably be "....stained with InstantBlue Coomassie stain...."

Thank you for catching this typo – corrected.

Reviewer #2 (Remarks to the Author):

This manuscript by Sharma and colleagues defines, at the molecular level, the first steps of maturation and assembly of the mitochondrial respiratory chain complex II and TCA cycle enzyme succinate dehydrogenase. The complex is formed by four subunits and the manuscript focuses on the maturation and assembly of subunit SDHA, which contains an FAD co-factor. Three intermediates are identified here, containing SDHA and the assembly factor SDHAF2, SDHA-SDHAF2 and the assembly factor SDHAF4, and SDHA-SDHAF4. The binding of the assembly factors to SDHA involves a change in disorder, which also facilitates the displacement of AF2 by AF4. In cellulo, and in vitro studies are well combined to demonstrate the order of events. The discussion is rich in finding parallelisms with other assembly systems, in which changes in disorder in assembly factors appear to be a common theme. Therefore, the conclusions have a broader impact in multiple biological systems. The manuscript is technically and conceptually sound and in the opinion of this reviewer should be published as it is.

We thank the reviewer for their support of this work.

Reviewer #3 (Remarks to the Author):

In their manuscript Sharma et al. describe how one of the subunits, SDHA in the mitochondrial CII complex is matured. The CII complex is essential for cell respiration and metabolism, and its misassembly is associated with several diseases, including ischemia and cancers. The authors have used pull-down assays, western blots and biophysical methods, including structure determination by x-ray crystallography to elucidate the order of the events in maturation of SDHA.

Although the manuscript is well written and the experimental work is solid, some clarity in the take home message of the work is missing. I am thus not sure if the authors main conclusion is the sequence of maturation events presented in Figure 6, which I find needs further justification (vide infra) - or if it is how disorder in SHDAF2 and SHDAF4 is important for this and other assembly processes as discussed on pages 8 - 9 in the manuscript. It thus also becomes unclear exactly how the authors see their results advance the field.

We thank the reviewer for their thoughtful consideration of the manuscript. Globally, we made modifications that we believe address all the reviewer's concerns. This revision included making major modifications within the abstract and introduction, adding and expanding information relevant to the sequence of events, modifying figure legends to include more details that support the mechanism, changing the abstract significantly, modifying the title and content of the Figure 7 (original Figure 6) legend to more accurately reflect the contents, and adding more information to the NMR, including additional SI figure panels. Details of each of these revisions are below.

In addition to the overall criticism, I have the following more specific points.

1. On the cover page it is confusing that the addresses are not listed in numerical order.

We thank the reviewer for pointing this out. In the revised, we modified the addresses so that departments from the same university were listed separately rather than being listed together.

2. p2, last sentence in the abstract. Is this the most important insight from the work and where the authors see its largest impact? What about the biological insight that understanding the assembly process give?? This is not mentioned here and is also largely missing from the discussion section fo the manuscript.

We thank the reviewer for pointing out that the take-home was not fully clear. To address this, we significantly modified the abstract to make it more apparent that these findings explain how intrinsic disorder regulates CII activity at the level of assembly.

3. p3, first paragraph of the results section. This is mainly background information and should be put in the introduction.

Per the reviewer's request, we moved this paragraph into the introduction.

4. p4, first paragraph and Figure 2a. Is only SDHA loaded with FAD tested in the pull-downs? How will the interaction pattern change if no FAD is present, if FAD is covalently bound and if dicarboxylate is added? Although some of this is mentioned in the previous paragraph it would be great to have these effects shown in the same pull-down assay where the other interactions are studied. It is also very hard to read the gel in Figure 2A as no labelling of the lanes are put in the figure.

We appreciate the reviewer identifying ways to improve the in vitro assay. In response to these comments, we fully redesigned this experiment to make it more systematic. The revised experiment allows the reader to step through the SDHA maturation process more logically and better explains the status of SDHA induced by each condition. Per this reviewer's request, this revised experiment includes dicarboxylate as a variable and includes more detailed labeling. We now show input for each gel. The associated display items now involve a four-panel figure, and this changes the organization of the main text figures. We clarified in the legend that the gel shown in the figure was representative and that a minimum of four replicates was performed with protein produced from independent preparations. We further indicated the exact number of replicates for each experiment in the legend.

SDHAF2 and SDHAF4 were used in 2-3x molar excess of SDHA, as indicated in the methods/figure legend. The input demonstrates the protein mixture before the Ni-NTA pull down.

5. p4, first paragraph. The authors should specifically mention which previous results in the literature that supports that SDHAF2 is the first factor to bind. That SDHAF2 has a lower Kd than SDHAF4 as the pull-down assay suggests does not necessarily mean that SDHAF2 binds first. E.g. the cellular concentrations could favour SDHAF4.

We thank the reviewer for this comment. We agree that affinity does not necessarily mean that a protein binds first in cells. We have now expanded the sentence to include the specific findings of past work and provide more context for why we believe that SDHAF2 binds first. In short, past work showed that SDHAF2 enhances the covalent flavinylation of SDHA and that SDHAF4 binds to flavinylated SDHA in cells. Combined with the stronger binding of SDHAF2 for apo-SDHA that we observe in vitro, this suggests that SDHAF2 is more likely to bind first during productive SDHA maturation. Our new data (Fig. 2) clearly demonstrate that SDHAF2 binds tightly to SDHA independent of flavinylation status, while robust binding of SDHAF4 was observed only to the holo-SDHA in a complex with SDHAF2. In the revised manuscript, we also more clearly explained the logic of sequential assembly factor action.

6. p4, last paragraph. From the native gels from the KO cells the authors conclude that SDHA-AF4 is not forming in cells. How is this consistent with the pull-down assays where both AF2 and AF4 form complexes with SDHA?

We thank the reviewer for this thoughtful comment, where the true response is relatively deep in nature. In fact, it is an area of active research. There are several reasons why this would be so. With the reviewer's grace, a full explanation of why we observe this association in vitro requires some initial and long explanation of preferred versus non-preferred protein maturation pathways. There are likely both preferred and non-preferred pathways of maturation in cells for Complex II (as well as any protein in the cell). Non-preferred pathways of maturation are unlikely to be fully excluded but may not be observable in cells when the preferred pathway is functioning correctly. These non-preferred pathways and interactions might only appear when concentrations of some components are greatly increased, as would be observed in the in vitro assays.

The existence of less preferred pathways of protein maturation is, in fact, predicted by using a chemical approach and mass action analysis. For the SDHA subunit of complex II, this mass-action diagram is relatively complex (see figure R2, located above in the response to reviewer 1). The presence of non-preferred pathways of maturation during SDHA maturation (or the maturation of any protein) explains a number of findings in the literature that seemed to contradict. For example, minor amounts of Complex II can assemble when SDHAF4 is knocked out in model organisms. In addition, SDHAF2 enhances covalent flavin attachment, but some cancer cells are able to assemble Complex II with covalent FAD following the knockout of SDHAF2.

While a full discussion of alternative mechanisms of assembly under disease conditions is outside of the scope of this work, we are actively in the process of collecting data that uses a computational approach in conjunction with the mass action diagram. The goal is to develop a predictive and quantitative model for Complex II assembly under normal versus perturbed cellular conditions. Preliminary data toward the investigations on the non-preferred pathways of maturation are relevant to the reviewer's query. For example, to identify which non-preferred pathways of Complex II maturation can be accessed under highly perturbed conditions, we overexpressed SDHAF4 in the SDHAF2 KO cell line, as requested. We still could not confidently identify this association in cells (see Figure R3, in the response to reviewer 1, above), although a small amount of this complex could be masked by the background from the antibody. This may be due to other conditions that are required to bind SDHAF4 with SDHA in the absence of SDHAF2, such as specific dicarboxylates at high concentrations. We are in the process of identifying what conditions would promote that interaction in cells. As another part of this future manuscript, we show that SDHAF4 knockout appears to result in a lower percentage of SDHA containing covalent flavin, even though SDHAF4 is not involved in covalent flavinylation either directly or indirectly. Instead, each of these results suggests that SDHA maturation is proceeding via a non-preferred pathway when cellular conditions are sufficiently perturbed and that this results in less-preferred protein associations. These types of situations might occur physiologically in a patient with a homozygous knockout of SDHAF2 or SDHAF4. We believe that we captured this non-preferred association in vitro by using concentrations that drove the association equilibrium toward a less-favored complex.

To improve the manuscript, we modified the assay shown in Figure 2a, as described in response to point 1, above. First, we ran additional controls that lay out the experiments more logically. In addition, we recognized that the mass action diagram predicts a concentration-dependence of SDHAF4 bindings to SDHA, i.e. what we would predict to be a non-preferred but non-excluded interaction in cells. Based on this, we adjusted the concentration levels of the assay such that a lower level of SDHAF4 binds (still not zero) under the conditions tested, thus sampling the non-preferred pathway less frequently. The revised manuscript includes this clearer assay but makes it clear that we do observe SDHA binding to SDHAF4 at a non-zero level.

Given the focus of this manuscript, we did not add a deep delve into preferred versus non-preferred pathways to the discussion. Instead, we plan to complete a future manuscript that rigorously tests how Complex II can mature under non-preferred conditions during disease. Some of the data in this response to review will be included in that manuscript.

7. p5, last paragraph. The authors mention CD spectra, which they also show in Fig S4. How these experiments were made is not written in the methods section. The CD spectrum in Figure S4b will be dominated by the folded SDHA that gives a strong signal. A spectrum of apo-SDHA should be subtracted to assess the structure of SDHAF4 in the complex!

We thank the reviewer for their attention to detail. In the revised manuscript, we included methods for the CD spectroscopy, which was unintentionally omitted from the initial submission.

In terms of the reviewer's concern that the folded protein would dominate the spectrum – we agree. We had included this spectrum not to show the folding of SDHAF2 but as a control to demonstrate to a reader outside the field what a typical CD spectrum for a folded protein looks like. In the revised SI legend, we now clarified that the rationale for this figure panel was to show a spectrum for a typical folded protein.

8. p5, last paragraph. The authors write that they have made a chemical shift analysis. I can't find where this is reported. It would be highly interesting to see if there is any low populated transient structure. Also in the 15N HSQC, there is a set of low intensity peaks suggesting that SDHAF4 is also found in an alternative conformation - and thus not fully disordered? Or is this an impurity? Showing SEC or SDS-PAGE of the sample would be appropriate.

The CS analysis does show varying flexibility throughout the peptide, but no sustained stretch of high stability and no secondary structure element prediction. As for the low-intensity peaks, they likely are not due to other proteins given their size and the sample purity. Instead, we interpret that these peaks are part of some metastable conformation(s) or slowly interchanging peaks between two forms.

To clarify this for readers, we now include three additional panels as a part of the SI Figure (now Figure S5). This includes the SEC chromatogram and the SDS-PAGE (Figures S5c and S5d). We also show the Chemical Shift Index analysis as calculated with the program Talos+ predicting secondary structure, and the calculated order parameter S2 indicating residue flexibility (Figure S5g). Neither shows any sign of secondary structure elements.

9. p6, section "SDHA-AF4 is the Third Protein-Protein Complex in SDHA Maturation" The SDHA:SDHAF4 complex for the crystal structure is prepared by adding excess SDHAF4 relative to SDHAF2. However, in the model presented in Figure 6 there is no concentration dependence going from the third to fourth state. So what is the driving force for releasing SDHAF2 from SDHA? The most obvious suggestion is that SDHAF2 and SDHAF4 interact with each other and that excess of either of these components will drive the other component out of the complex with SDHA, by forming a SDHAF2:SDHAF4 complex. This should be tested.

We thank the reviewer for thinking deeply about the mechanisms. In this case, treatment of the SDHA sample with an excess of its binding partner was performed to ensure that the sample was saturated prior to crystallization; this is a typical procedure in the field to improve homogeneity and is not meant to imply anything about the mechanism.

Given the reviewer's curiosity as to whether SDHAF2 and SDHAF4 interact directly, we performed an immunoprecipitation between these two assembly factors and show that there is no detectable direct interaction. This panel is now included as SI Figure 1.

10. p7, first line. The authors write "To test whether *intrinsic disorder* is important for the transfer...". From their experiments what the authors are testing is whether the disordered regions (not disorder in itself) are important for the interaction. They are not testing if any disordered sequence is working or if it is possible to replace the disordered regions with ordered regions. At the end of the second paragraph on p7 the authors correctly conclude that "These findings implicate the *intrinsically disordered regions* in the process..."

We thank the reviewer for their close attention to detail. We edited the manuscript throughout with a specific focus on revising this wording to be more precise. The reviewer is correct that the deletion mutagenesis implicates the region of the protein that is intrinsically disordered.

11. p7, second paragraph of the discussion. The authors write “We show that SDHA proceeds through at least three metastable intermediates (Figure 6) prior to being incorporated into CII...” The authors have not shown the incorporation of SDHA into CII.

We thank the reviewer for their close attention to detail. We agree that this language would benefit from improved precision. In response, we modified the sentence to say that “...SDHA proceeds through at least three metastable intermediates (Figure 6) during its maturation.” We further modified the legend of this figure (now Figure 7) to add “The handling steps of SDHA following this initial maturation process but prior to integration into functional CII (PDB ID 3SFD) remain unknown at this time.”

12. Figure 2. Labels are missing on the lanes of the gel in panel a. The figure legend for panel b report that the gels are representative of three experiments. In Fig S1a that shows the quantification there are 4-5 data points on each bar. The same goes for panel b, where there are 3-4 points in Fig S1b-h.

In response to comments from both R1 and R3 on the clarity of Figure 2a, we repeated this experiment under a broader range of conditions, more systematically, and with additional controls, as described above. As a part of this, we moved quantitation to the main figure, where N=4.

In addition, we modified the legend to the former S1b-h (now Fig. S2 b-h) to be clearer that each point represents one replicate and to explicitly indicate the number of replicates in the legend.

13. Figure 3. It is almost impossible to see the FAD in panel a, although the legend to panel a describes how FAD is displaced. In panel d, the labels on the lanes of the gel are missing.

We thank the reviewer for noticing this. To improve the clarity of Figure 3a (now Figure 4a), we added a box around the FAD and indicated how this relates to the inset in panel c.

14. The authors should carefully proofread the manuscript, as there are several missing words throughout the text.

We appreciate this attention to detail. We did a careful additional read of the manuscript and SI, which indeed identified a number of typos and missing words. We appreciate your calling our attention to this. If the reviewer finds additional items that we missed, we would appreciate a specific list.

REVIEWER COMMENTS

Reviewer #1 (Remarks to the Author):

In this revised manuscript by Sharma et al, entitled “Disordered-to-ordered transitions in assembly factors allow the Complex II catalytic subunit to switch binding partners” the authors have addressed the multiple concerns raised. The main text is much more accessible to the reader and is better supported by the data. The manuscript has benefited from the inclusion of the redesigned and expanded data presented in Figure 2 which proved to be more informative than the original data and better supports the conclusions.

Reproducibility and critical evaluation of the data by the journal readership will now be possible with the improved data labelling, correction of method omissions and errors and the additional author explanations.

Minor corrections

Labelling error in Figure 3b (Lanes 5 and 7) and in the corresponding quantitation graphs:
“...../SDHAFrec” should be “...SDHAF4rec”, “...../SDHAF0E” should be “...SDHAF4OE”

Typographical error on page 9: “migrateed” should be “migrated”.

Reviewer #3 (Remarks to the Author):

In their revised manuscript Sharma et al. have addressed most of my and the other reviewer’s points of criticism well. I find that the revised manuscript is much improved - both in the analysis of the data and in the discussions in the text.

I still have one issue that the authors need to address. In their analysis of the chemical shift data from NMR they use talos+ to assess if there is transient structure in SDHAF4. Talos+ is made to find the most likely dihedral angles in structured proteins and not for assessing transient structure. The authors should use a method that is designed for analysing transient secondary structures. I will recommend POTENCI (<https://st-protein02.chem.au.dk/potenci/> doi: 10.1007/s10858-018-0166-5)

Reviewer #1 (Remarks to the Author):

In this revised manuscript by Sharma et al, entitled “Disordered-to-ordered transitions in assembly factors allow the Complex II catalytic subunit to switch binding partners” the authors have addressed the multiple concerns raised. The main text is much more accessible to the reader and is better supported by the data. The manuscript has benefited from the inclusion of the redesigned and expanded data presented in Figure 2 which proved to be more informative than the original data and better supports the conclusions.

Reproducibility and critical evaluation of the data by the journal readership will now be possible with the improved data labelling, correction of method omissions and errors and the additional author explanations.

We thank the reviewer for their support of the revised manuscript.

Minor corrections

Labelling error in Figure 3b (Lanes 5 and 7) and in the corresponding quantitation graphs: “...../SDHAFrec” should be “...SDHAF4rec”, “...../SDHAFOE” should be “...SDHAF4OE”

Typographical error on page 9: “migrateed” should be “migrated”.

We thank the reviewer for pointing this out. It is corrected in the revised version.

Reviewer#3 (Remarks to the Author):

In their revised manuscript Sharma et al. have addressed most of my and the other reviewer’s points of criticism well. I find that the revised manuscript is much improved - both in the analysis of the data and in the discussions in the text.

We thank reviewer 3 for support of the revised manuscript.

I still have one issue that the authors need to address. In their analysis of the chemical shift data from NMR they use talos+ to assess if there is transient structure in SDHAF4. Talos+ is made to find the most likely dihedral angles in structured proteins and not for assessing transient structure. The authors should use a method that is designed for analysing transient secondary structures. I will recommend POTENCI (<https://st-protein02.chem.au.dk/potenci/> doi: 10.1007/s10858-018-0166-5

We thank the reviewer for opening a discussion of the best tools for analyzing NMR data of intrinsically disordered proteins. We also appreciate the recommendation of the POTENCI¹ tool. As suggested by the reviewer we used POTENCI, as well as two other tools; chemical shift secondary structure population inference (CheSPI)² and Poulsen³, known for NMR secondary chemical analysis. All these methods supported the Talos+ analysis and confirm that human SDHAF4 is an intrinsically disordered protein. The results from CheSPI are

now included in the updated supplementary information as figure S5h. We note that the conclusion that SDHAF4 is an intrinsically disordered protein in isolation is also supported by Circular Dichroism spectroscopy and is not dependent on the method of calculating the NMR reference shifts.

References:

- 1 Nielsen, J. T. & Mulder, F. A. A. POTENCI: prediction of temperature, neighbor and pH-corrected chemical shifts for intrinsically disordered proteins. *J Biomol NMR* **70**, 141-165, doi:10.1007/s10858-018-0166-5 (2018).
- 2 Nielsen, J. T. & Mulder, F. A. A. CheSPI: chemical shift secondary structure population inference. *J Biomol NMR* **75**, 273-291, doi:10.1007/s10858-021-00374-w (2021).
- 3 Kjaergaard, M. & Poulsen, F. M. Sequence correction of random coil chemical shifts: correlation between neighbor correction factors and changes in the Ramachandran distribution. *J Biomol NMR* **50**, 157-165, doi:10.1007/s10858-011-9508-2 (2011).

REVIEWERS' COMMENTS

Reviewer #3 (Remarks to the Author):

I'm fine with the analysis of the chemical shifts. However, the authors write on line 634: The first program, Talos+ (Figure S5g), is validated for use on intrinsically disordered proteins⁶⁵. This statement is surprising to me and I can't find support for it in reference 65. I suggest that the authors delete this sentence and the Talos+ figure (or at least the lower part of the figure).

Reviewer #3 (Remarks to the Author):

I'm fine with the analysis of the chemical shifts. However, the authors write on line 634: The first program, Talos+ (Figure S5g), is validated for use on intrinsically disordered proteins⁶⁵. This statement is surprising to me and I can't find support for it in reference 65. I suggest that the authors delete this sentence and the Talos+ figure (or at least the lower part of the figure).

We used Talos+ as one of three orthogonal methods to analyze the data NMR data. We modified the text to remove the phrase that Talos+ was validated on IDPs. We needed to retain the reference itself because Talos+ was used as one of three methods to analyze the data. We also added a note to the author checklist.